# Inhibition of proteasome rescues a pathogenic variant of respiratory chain assembly factor COA7

Karthik Mohanraj[1,2,3], Michal Wasilewski[1,3,*] (ID), Cristiane Benincá[4], Dominik Cysewski[5], Jaroslaw Poznanski[6], Paulina Sakowska[3], Zaneta Bugajska[1], Markus Deckers[7], Sven Dennerlein[7], Erika Fernandez-Vizarra[4] (ID), Peter Rehling[7,8], Michal Dadlez[5], Massimo Zeviani[4] (ID) & Agnieszka Chacinska[1,2,3,**] (ID)

## Abstract

Nuclear and mitochondrial genome mutations lead to various mitochondrial diseases, many of which affect the mitochondrial respiratory chain. The proteome of the intermembrane space (IMS) of mitochondria consists of several important assembly factors that participate in the biogenesis of mitochondrial respiratory chain complexes. The present study comprehensively analyzed a recently identified IMS protein cytochrome *c* oxidase assembly factor 7 (COA7), or REScpiratory chain Assembly 1 (RESA1) factor that is associated with a rare form of mitochondrial leukoencephalopathy and complex IV deficiency. We found that COA7 requires the mitochondrial IMS import and assembly (MIA) pathway for efficient accumulation in the IMS. We also found that pathogenic mutant versions of COA7 are imported slower than the wild-type protein, and mislocalized proteins are degraded in the cytosol by the proteasome. Interestingly, proteasome inhibition rescued both the mitochondrial localization of COA7 and complex IV activity in patient-derived fibroblasts. We propose proteasome inhibition as a novel therapeutic approach for a broad range of mitochondrial pathologies associated with the decreased levels of mitochondrial proteins.

**Keywords** COA7/RESA1; mitochondrial disease; proteasome; protein degradation; protein import
**Subject Categories** Genetics, Gene Therapy & Genetic Disease; Pharmacology & Drug Discovery

See also: **M Habich & J Riemer** (May 2019)

## Introduction

The constant import of nuclear-encoded mitochondrial proteins into mitochondria is required for healthy and functional mitochondria. Mitochondrial precursor proteins utilize various versatile import machineries for efficient import and proper localization into the destined subcompartments, including the outer membrane (OM), intermembrane space (IMS), inner membrane (IM), and matrix (Neupert & Herrmann, 2007; Chacinska *et al*, 2009; Endo & Yamano, 2010; Schmidt *et al*, 2010; Schulz *et al*, 2015; Wasilewski *et al*, 2017). The mitochondrial intermembrane space import and assembly (MIA) pathway is responsible for the import and stable accumulation of cysteine-rich proteins in the IMS (Chacinska *et al*, 2004; Naoe *et al*, 2004; Terziyska *et al*, 2005). The MIA pathway includes oxidoreductase MIA40/CHCHD4 (Mia40 in yeast), which recognizes and oxidizes substrate proteins, and ALR/GFER (Erv1 in yeast), which receives electrons from oxidation of a substrate (Hofmann *et al*, 2005; Mesecke *et al*, 2005; Muller *et al*, 2008; Banci *et al*, 2009, 2010; Sztolsztener *et al*, 2013; Koch & Schmid, 2014; Peleh *et al*, 2016). The classic MIA40 substrates are mostly small proteins (< 20 kDa), such as TIMM8A and COX6B, with a specific arrangement of cysteine residues, such as $CX_3C$ or $CX_9C$ (Koehler, 2004; Milenkovic *et al*, 2009; Sideris *et al*, 2009; Bourens *et al*, 2012; Fischer *et al*, 2013). MIA40 is also involved in the import of non-canonical substrates, which do not possess $CX_3C$ or $CX_9C$ signals in their sequence such as apurinic/apyrimidinic endonuclease (APE1), cellular tumor antigen p53, and mitochondrial calcium uptake 1 (MICU1) in metazoan and Tim22 in yeast (Wrobel *et al*, 2013; Zhuang *et al*, 2013; Barchiesi *et al*, 2015; Petrungaro *et al*, 2015).

Mitochondrial dysfunction is associated with many diseases (Costa & Scorrano, 2012; Viscomi *et al*, 2015; Gorman *et al*, 2016;

1 Laboratory of Mitochondrial Biogenesis, Centre of New Technologies, University of Warsaw, Warsaw, Poland
2 ReMedy International Research Agenda Unit, Centre of New Technologies, University of Warsaw, Warsaw, Poland
3 Laboratory of Mitochondrial Biogenesis, International Institute of Molecular and Cell Biology, Warsaw, Poland
4 MRC Mitochondrial Biology Unit, University of Cambridge, Cambridge, UK
5 Mass Spectrometry Lab, Department of Biophysics, Institute of Biochemistry and Biophysics, Warsaw, Poland
6 Department of Biophysics, Institute of Biochemistry and Biophysics, Warsaw, Poland
7 Department of Cellular Biochemistry, University of Göttingen, Göttingen, Germany
8 Max Planck Institute for Biophysical Chemistry, Göttingen, Germany
*Corresponding author. Tel: +48 22 55 43826; E-mail: m.wasilewski@cent.uw.edu.pl
**Corresponding author. Tel: +48 22 55 43639; E-mail: a.chacinska@cent.uw.edu.pl

Suomalainen & Battersby, 2018). Among these, the impaired biogenesis of MIA40 and its substrates contributes to a significant percentage of mitochondrial pathology (Koehler *et al*, 1999; Tranebjaerg *et al*, 2000; Roesch *et al*, 2002; Friederich *et al*, 2017; Erdogan *et al*, 2018). Interestingly, mutations of MIA40 substrates (e.g., TIMM8A, COX6B, and NDUFB10) are associated with the significant loss of proteins in patients, consequently leading to impairments of the biogenesis of many other IMS and IM proteins.

Many mitochondrial precursor proteins, before they productively reach the mitochondrial compartment, are under control of the ubiquitin–proteasome system (UPS), a major protein-degrading pathway that is involved in maintaining cellular protein homeostasis (Radke *et al*, 2008; Bragoszewski *et al*, 2013). Furthermore, IMS proteins can undergo reductive unfolding and slide back to the cytosol where they are also degraded by the UPS (Bragoszewski *et al*, 2015). However, the contribution of cytosolic protein control mechanisms to mitochondriopathies has not been investigated.

The present study identified a cysteine-rich IMS protein, COA7, as a new non-canonical substrate of MIA40. COA7 is a Sel1 repeat-containing protein that was previously shown to be involved in the assembly of respiratory chain complexes I and IV (Kozjak-Pavlovic *et al*, 2014; Martinez Lyons *et al*, 2016). We characterized the interaction of COA7 with MIA40 and its import and localization in mitochondria. We also characterized pathogenic disease-causing COA7 mutants (Martinez Lyons *et al*, 2016) as defective in their biogenesis and degraded in the cytosol by the UPS. Importantly, inhibition of the UPS system led to the partial rescue of defective COA7 variants, thus suggesting a conceptually novel strategy for therapeutic interventions.

# Results

### COA7, new MIA40-interacting protein

To identify new interacting partners/substrates of MIA40, we performed affinity purification of MIA40$_{FLAG}$ that was induced in human Flp-In T-Rex 293 cells and subjected the eluate fraction to mass spectrometry. Specificity was calculated as the ratio of the protein signal intensity in the bait purification (MIA40$_{FLAG}$) to the protein signal intensity in the control purification. The identification of previously known MIA40-interacting proteins (ALR and AIF1) and MIA40 substrates (TIMM13 and COX6B) validated the specificity of our affinity purification. Among the proteins that co-purified with MIA40$_{FLAG}$, we identified a cysteine-rich protein COA7 with high abundance and specificity (Fig 1A, Dataset EV1). COA7 is very distinct from classic MIA40 substrates due to the presence of 13 cysteine residues that are neither surrounded by classic MIA40-targeting signals like mitochondria IMS-sorting signal (MISS)/IMS-targeting signal (ITS) (MISS/ITS) nor arranged in classic motifs (CX$_3$C or CX$_9$C) (Milenkovic *et al*, 2009; Sideris *et al*, 2009). Immunoblotting of the MIA40$_{FLAG}$ eluate fraction with COA7 antibody confirmed our mass spectrometry readout (Fig 1B, lane 4). As a positive control, we observed co-purification of ALR and AIF in the eluate fraction (Fig 1B, lane 4) as well as weak signal of transiently interacting substrates of MIA40 such as TIMM10B and TIMM13 (Fig EV1A). To verify the possible interaction between COA7 and ALR (a known MIA40 partner), we performed affinity

purification via ALR$_{FLAG}$, and only a small amount of COA7 co-purified with ALR (Fig EV1B).

### COA7 interacts with MIA40 through disulfide bonds

Since MIA40 interacts with many precursor proteins via disulfide bonds, we investigated whether the interaction with COA7 involves the formation of disulfide bridges. We first performed affinity purification via MIA40$_{FLAG}$ and probed the eluate fraction using MIA40 and COA7 antibodies under reducing (dithiothreitol [DTT]) and non-reducing (iodoacetamide [IAA]) conditions. Immunoblotting against both COA7 and MIA40 revealed a band of approximately 45 kDa (Fig 1C, lanes 2 and 3) under non-reducing conditions, which would match the combined molecular weight of a covalent, disulfide-bonded complex of MIA40$_{FLAG}$ (15.9 kDa) and COA7 (25.7 kDa). Therefore, we assumed that COA7 may interact with MIA40 through disulfide bonds. We tested this hypothesis using mitochondria that were isolated from cells induced to express wild-type and cysteine residue mutant versions of MIA40$_{FLAG}$, namely C53S, C55S, and C53,55S (denoted SPS), under non-reducing conditions (Fig EV1C). We observed two bands that represented native MIA40-COA7 and MIA40$_{FLAG}$-COA7 complexes in cells that expressed wild-type and C53S MIA40$_{FLAG}$ (Fig EV1C, lanes 2, 3, 7, and 8) and only one band that corresponded to the native MIA40-COA7 complex in cells that expressed C55S MIA40$_{FLAG}$ and SPS MIA40$_{FLAG}$, respectively (Fig EV1C, lanes 4, 5, 9, and 10). We also confirmed this observation in cellular protein extract that was probed with anti-COA7 antibody. This indicated the existence of a covalently bound complex between MIA40 and COA7 under *in vivo* conditions (Fig EV1D). The second cysteine residue in the CPC motif of MIA40 was previously shown to be crucial for the interaction with classic MIA40 substrates (Banci *et al*, 2009). Therefore, we checked the cysteine dependency of COA7 binding by performing affinity purification via MIA40 cysteine mutant protein (C53S, C55S, and SPS) that was induced in Flp-In T-REx 293 cells. We observed COA7 only in the eluate fractions of wt MIA40$_{FLAG}$ and C53S MIA40$_{FLAG}$ (Fig 1D, lanes 6 and 7) and not in the other mutants (Fig 1D, lanes 9 and 10). Thus, the second cysteine of the CPC motif (i.e., C55) was crucial for the disulfide bonding of MIA40 with COA7.

### COA7 exists as an oxidized protein in the IMS

COA7 contains 13 cysteine residues. Hence, we evaluated the phylogenic diversity of cysteine residues in COA7 orthologs in eukaryotes and found the strong conservation of ten residues (C28, C37, C62, C71, C110, C111, C142, C150, C182, and C190) across eukaryotes, whereas three other cysteine residues (C24, C95, and C172) were present only in vertebrates (Appendix Fig S1A). We hypothesized that the conserved cysteine residues may be involved in intramolecular disulfide bonds. To predict the cysteine residues that are involved in the disulfide bonds, we modeled the structure of COA7 based on three proteins with high sequence similarity: Helicobacter cysteine-rich protein C (1OUV), *Helicobacter pylori* cysteine-rich protein B (4BWR), and a protective antigen that is present in *Escherichia coli* (1KLX). The modeled structure of COA7 possessed five disulfide bonds with a unique pattern (disulfide bridges between cysteine residues that were separated by nine or eleven amino

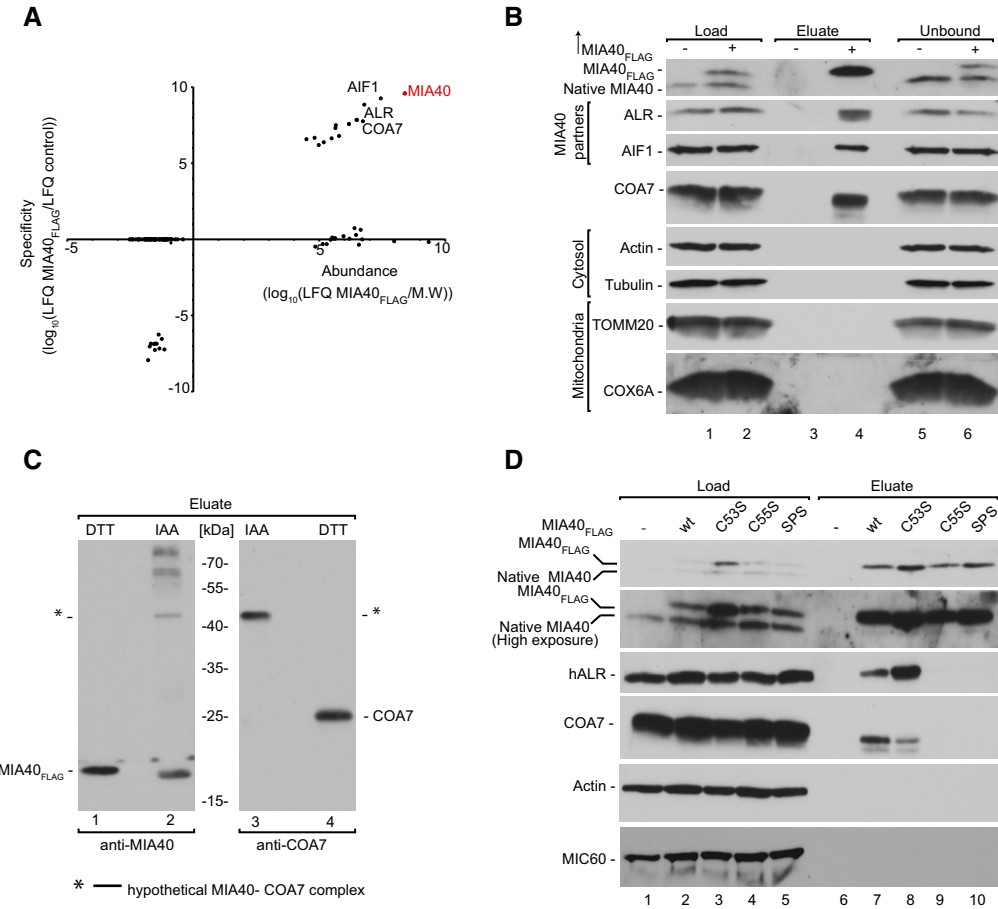

**Figure 1. COA7 interacts with MIA40 by disulfide bonding.**

A   Protein abundance is the normalized signal intensity (LFQ) for a protein divided by its molecular weight. Specificity (enrichment) is the ratio of the protein LFQ intensity in the MIA40$_{FLAG}$ fraction to control samples. The LFQ for proteins that were not detected in the control samples was arbitrarily set to 1 for calculation purposes. LFQ, label-free quantification; M.W., molecular weight.

B   Flp-In T-REx 293 cells induced to express MIA40$_{FLAG}$ were solubilized, and the affinity purification of MIA40$_{FLAG}$ was performed. Fractions were analyzed by SDS–PAGE and Western blot. Load: 2.5%. Eluate: 100%. Unbound: 2.5%.

C   Flp-In T-REx 293 cells induced to express MIA40$_{FLAG}$ were solubilized, and the affinity purification of MIA40$_{FLAG}$ was performed. Eluate fractions were solubilized under reducing (DTT) or non-reducing (IAA) conditions and analyzed by SDS–PAGE and Western blot. DTT, dithiothreitol; IAA, iodoacetamide. Eluate: 100%.

D   Cellular protein extracts from Flp-In T-REx 293 cells with induced expression of wild-type or mutant MIA40$_{FLAG}$ were subjected to affinity purification. Load and eluate fractions were analyzed by reducing SDS–PAGE and Western blot. Load: 2.5%. Eluate: 100%.

Source data are available online for this figure.

acids). The structure resembled those of the template proteins with subsequent helix-turn-helix (HTH) motifs that formed a super-helical structure that was analogous to previously reported structures. The N-terminal part of the protein did not have a proper template. Therefore, we modeled it *de novo* and manually adjusted the improperly modeled C24–C37 disulfide bridge to C28–C37. This modification was based on the following: (i) It fit the evolutionary conservation of cysteine residues, and (ii) formation of the C28–C37 bond concurs with the pattern of other disulfide bonds (i.e., cysteine separated by nine or 11 amino acids) in the protein. Hence, we proposed that the remaining three cysteine residues (C24, C95, and C172) were expected to be in a reduced state (Appendix Fig S1B).

To validate our prediction of the COA7 redox state, we employed direct and indirect thiol trapping assays on the mitochondria that were isolated from human embryonic kidney 293

(HEK293) cells. The direct thiol trapping assay is based on the differential use of alkylating agents that bind free thiol residues in proteins and modify their migration in sodium dodecyl sulfate–polyacrylamide gel electrophoresis (SDS–PAGE). Binding of the low-molecular-mass agent IAA is neutral, whereas modification with 4-acetamido-4-maleimidylstilbene-2,2-disulfonic acid (AMS) alters the molecular mass of the protein by 0.5 kDa per each thiol residue. In the direct thiol assay, AMS modified the migration of COA7 compared with IAA by approximately 1.5 kDa, which corresponds to three free thiol residues (Fig 2A, lanes 2 and 3). Therefore, we presumed that ten cysteine residues would be involved in the disulfide bond, and the remaining three could be in a reduced state. To test this hypothesis, we performed an indirect thiol trapping assay that involved two-step thiol modification. Free thiols were first blocked with IAA, and the remaining cysteine residues

that formed disulfide bridges were reduced with DTT and modified with AMS. The observed shift corresponded to cysteine residues that formed disulfide bridges in native COA7 (Fig 2B, lane 4). As expected, we observed migration that corresponded to approximately ten cysteine residues. As a control, we treated mitochondria directly with DTT to completely reduce all native disulfide bonds and then modified them with AMS (Fig 2B, lane 5). We observed higher migration of COA7 compared with lane 4. These observations suggest that among the thirteen cysteine residues, likely ten are involved in disulfide bonds, and the other three exist in a reduced state. Based on the thiol trapping experiments and *in silico*

modeling, we propose that COA7 has five Sel-1 domain-like repeats that are stabilized by disulfide bridges. The Sel-1 domains are characterized by a specific arrangement of cysteine residues (i.e., the fourth amino acid and fifth amino acid from the last amino acid in each domain is always a cysteine).

To analyze the submitochondrial localization of COA7, we performed hypo-osmotic swelling of mitochondria that were isolated from HEK293 cells. In intact mitochondria, COA7 was preserved from degradation, similar to other proteins that were localized inside mitochondria, such as ALR and coproporphyrinogen oxidase (CPOX) (IMS), TIMM22, and COX6B (IM), and HSP60 and ATP5A

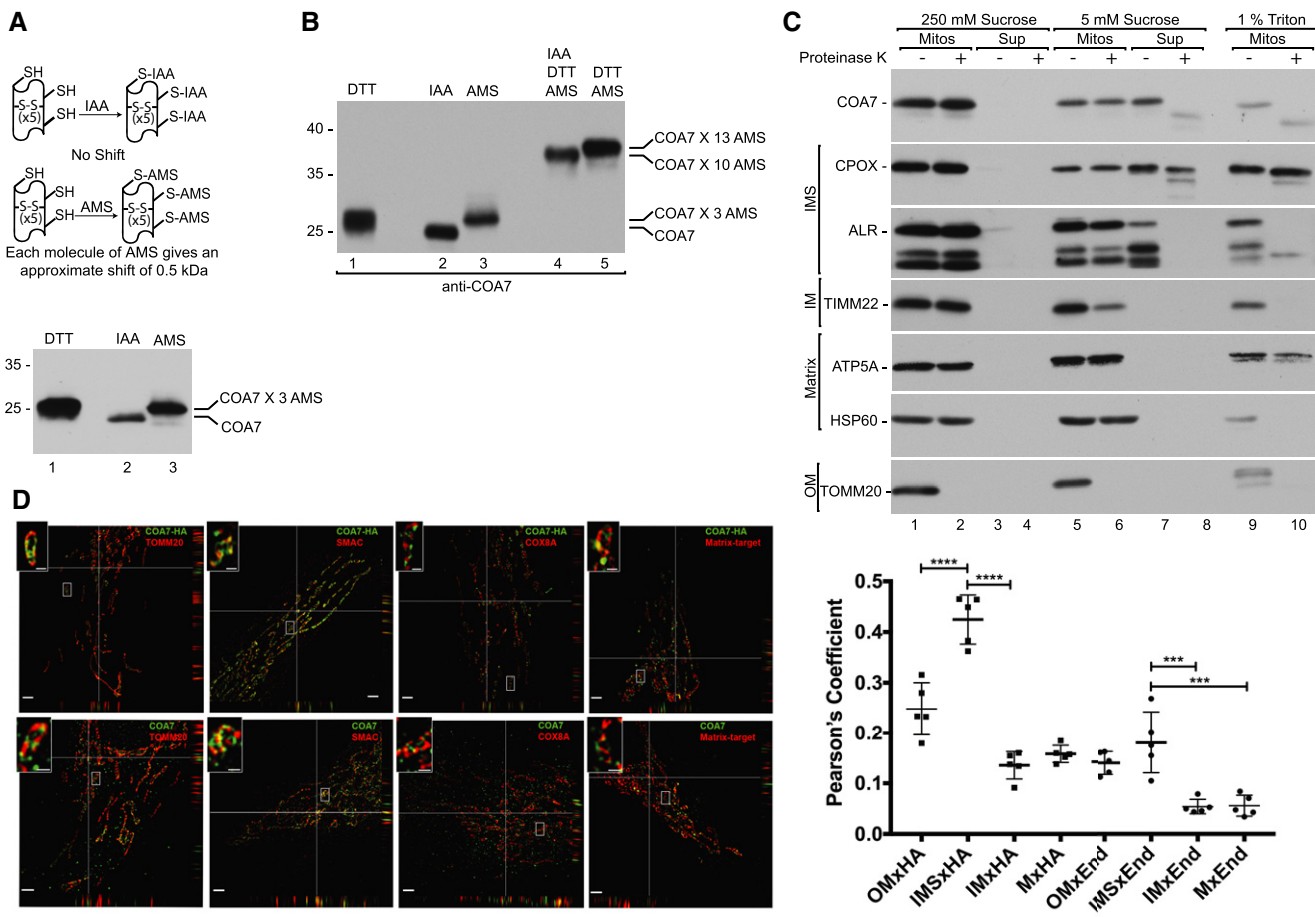

**Figure 2. COA7 is an oxidized protein in the intermembrane space of human mitochondria.**

A   Schematic representation of the thiol trapping assay. Mitochondria were solubilized in sample buffer with either dithiothreitol (DTT), iodoacetamide (IAA), or 4-acetamido-4-maleimidylstilbene-2,2-disulfonic acid (AMS). The samples were analyzed by SDS–PAGE and Western blot.

B   Indirect thiol trapping assay. Mitochondria were pretreated with IAA as indicated to block free cysteine residues, and disulfide bonds were subsequently reduced by DTT. Mitochondria were solubilized in sample buffer with AMS.

C   Localization of mitochondrial proteins analyzed by limited degradation by proteinase K in intact mitochondria (250 mM sucrose), mitoplasts (5 mM sucrose), and mitochondrial lysates (1% Triton X-100). The samples were analyzed by SDS–PAGE and Western blot. Mitos, mitochondria; Sup, post-mitochondria supernatant; OM, outer membrane; IM, inner membrane; IMS, intermembrane space.

D   N-SIM super-resolution micrographs of one *Z*-stack (0.15 μm) orthogonal section (*XYZ*) of HeLa cells or HeLa cells that stably expressed COA7-HA transfected with different markers TOMM20-DsRed (OM), COX8A-DsRed (IM), and matrix targeted photoactivatable GFP (Matrix target) labeled with anti-HA, anti-COA7, and Smac/Diablo (IMS) antibodies. The picture represents the majority population of cells from three independent experiments. Scale bar = 2 μm, scale bar in the magnified insert = 0.5 μm. The panel shows Pearson's coefficient in a co-localized volume of different subcompartment combinations with anti-HA or anti-COA7. The data are expressed as a mean ± SD (*n* = 5). ***$P < 0.001$ [IMSxEnd vs. IMxEnd $P = 0.0001$; IMSxEnd vs. MxEnd $P = 0.0002$], ****$P < 0.0001$ [OMxHA vs. IMSxHA $P < 0.0001$; IMSxHA vs. IMxHA $P < 0.0001$] (one-way ANOVA). M, matrix; End, endogenous COA7; HA, COA7-HA.

Source data are available online for this figure.

(matrix). As expected, the OM protein TOMM20 was efficiently degraded by proteinase K in intact mitochondria (Fig 2C, lanes 1–4). Upon rupturing the OM by swelling in hypotonic buffer, proteinase K degrades proteins that are exposed to the IMS, whereas matrix proteins remain protected. Accordingly, we observed the slight degradation of IM proteins (TIMM22) that faced the IMS side (Fig 2C, lanes 5–8). The matrix protein Hsp60 was completely resistant to proteinase K degradation, even upon swelling, and was degraded only by 1% Triton X-100 treatment (Fig 2C, lanes 9 and 10). Under these conditions, COA7 was present in the supernatant fraction similar to the soluble IMS proteins like CPOX and ALR and was sensitive to proteinase K (Fig 2C, lanes 5–8). COA7 has been previously reported to localize to the IMS or matrix (Kozjak-Pavlovic et al, 2014; Martinez Lyons et al, 2016). To further confirm the submitochondrial localization of COA7, we performed microscopy using the structured illumination technique (N-SIM). Several combinations of antibodies and targeted fluorescent proteins were used in order to verify the specificity of the observed co-localization. In particular, we targeted fluorophores to a single mitochondrial subcompartment and observed positive co-localization in the case of OM × OM (TOMM20 and TOMM20), IM × IM (SDHA and COX8A), and matrix × matrix (ACO2 and matrix target green fluorescent protein [GFP]) (Appendix Fig S1C). Conversely, we observed no co-localization when different subcompartments were stained: OM × IMS (TOMM20 and SMAC), OM × IM (SDHA and TOMM20), IM × matrix (COX8A and matrix target GFP), IMS × matrix (SDHA and MDH2), and OM × matrix (TOMM20 and matrix target GFP) (Appendix Fig S2C). These observations indicated that the technique provided sufficient resolution (~110 nm) to pair the correct subcompartment combinations of the positive controls TOMM20 × TOMM20 (OM × OM), COX8A × SDHA (IM × IM), and matrix target GFP × ACO2 (matrix × matrix) from all of the combinations that were used as negative controls. We then compared the signal of COA7 with the markers of different mitochondrial subcompartments. We observed an increase in co-localization only with the IMS marker SMAC and not with markers of other subcompartments (Fig 2D). The IMS localization of COA7 was similar with both endogenous COA7 and HA-tagged overexpressed COA7. Altogether, these findings indicate that COA7 is a soluble protein of the IMS (Fig 2D).

## COA7 does not influence the MIA pathway

The relatively strong interaction between COA7 and MIA40 (see Figs 1B and EV1A) prompted us to explore the potential role of COA7 in the MIA pathway. We first examined the steady-state levels of mitochondrial proteins upon COA7 overexpression and found no changes in the levels of MIA40 or its dependent proteins, including ALR, COX6B, and TIMM22 (Fig 3A). We next investigated the influence of COA7 on the import of MIA40 substrates by importing two substrates, TIMM8A ($CX_3C$) and COX19 ($CX_9C$), into mitochondria that have overexpressed levels of COA7. We did not observe any significant difference in the import of TIMM8A or COX19 compared with the control (Fig 3B). These observations suggest that COA7 overexpression does not influence either the steady-state levels of MIA40 or the import of MIA40 substrates.

Cysteine residues in the CPC motif of MIA40 are crucial for the recognition and import of MIA40 substrates. We previously established that the second cysteine in the CPC motif of MIA40 is crucial for the interaction with COA7 (Figs 1D and EV1C and D). Therefore, we evaluated the levels of free MIA40 that is available for the import of substrates under non-reducing conditions. As expected, COA7 overexpression did not significantly affect the level of free MIA40 (Fig 3C, lanes 2 and 4). We next checked the oxidation state of the cysteine residues in the CPC motif of MIA40 upon COA7 overexpression using the thiol-binding reagent PEG-PCMal, which induces a shift of approximately 5 kDa per cysteine residue. The specificity of the observed bands was confirmed in cells that had genetically lower levels of MIA40 (Appendix Fig S2A and B). We verified oxidation of the CPC motif of MIA40 in total cells and in mitochondria that were isolated from cells that transiently overexpressed COA7 (Fig 3D, Appendix Fig S2C and D). The treatment yielded two bands that were specific to MIA40. The lower band corresponded to the migration of MIA40 with PEG-PCMal that was bound to one cysteine residue (C4). The second band corresponded to the binding of PEG-PCMal to three cysteine residues (C4, C53, and C55). The CPC motif of MIA40 was oxidized to a similar extent in COA7-overexpressed and control samples although an overall level of oxidation was higher in isolated mitochondria than in total cells (Fig 3D, lanes 3 and 4, Appendix Fig S2D, lanes 1 and 2) which is consistent with partial oxidation of proteins during mitochondria isolation (Topf et al, 2018). Thus, we confirmed that COA7 overexpression did not affect the redox state of cysteine residues in the CPC motif, suggesting that both cysteine residues were equally available for importing MIA40 substrates, such as TIMM8A and COX19. This observation complemented our previous finding that COA7 overexpression did not affect the import of MIA40 substrates.

The role of ALR in the MIA pathway is well established, and the interaction between MIA40 and ALR is crucial for maintaining MIA40 activity (Banci et al, 2012; Sztolsztener et al, 2013). To investigate whether COA7 influences this crucial interaction, we performed affinity purification via $MIA40_{FLAG}$ to evaluate the MIA40-ALR interaction upon COA7 overexpression. The levels of ALR that co-purified with MIA40 were similar in cells that transiently overexpressed COA7 and control cells (Fig 3E, lanes 1 and 2). We did not observe any difference in the MIA40-ALR interaction under COA7-knockdown conditions (Fig EV2A). Consequently, COA7 knockdown using two different oligonucleotides did not alter the levels of MIA40 or its substrates, such as COX6B, TIMM9, and TIMM22 (Fig EV2B). We also examined the MIA40-ALR interaction by affinity purification via $ALR_{FLAG}$ under similar conditions. The amount of MIA40 that co-purified with ALR was similar under conditions of both COA7 knockdown and overexpression (Fig EV2C and D). Thus, we conclude that COA7 neither influences MIA pathway activity nor the import of MIA40 substrates to the IMS.

## MIA40 facilitates the import of COA7 into mitochondria

To investigate whether MIA40 is involved in the import of COA7, we performed *in organello* import of radiolabeled [$^{35}$S]COA7 precursor into mitochondria that were isolated from cells that were induced to express $MIA40_{FLAG}$. We observed a significant increase in COA7 import in mitochondria with $MIA40_{FLAG}$ compared with the control (Fig 4A). As a positive control, import of the classic

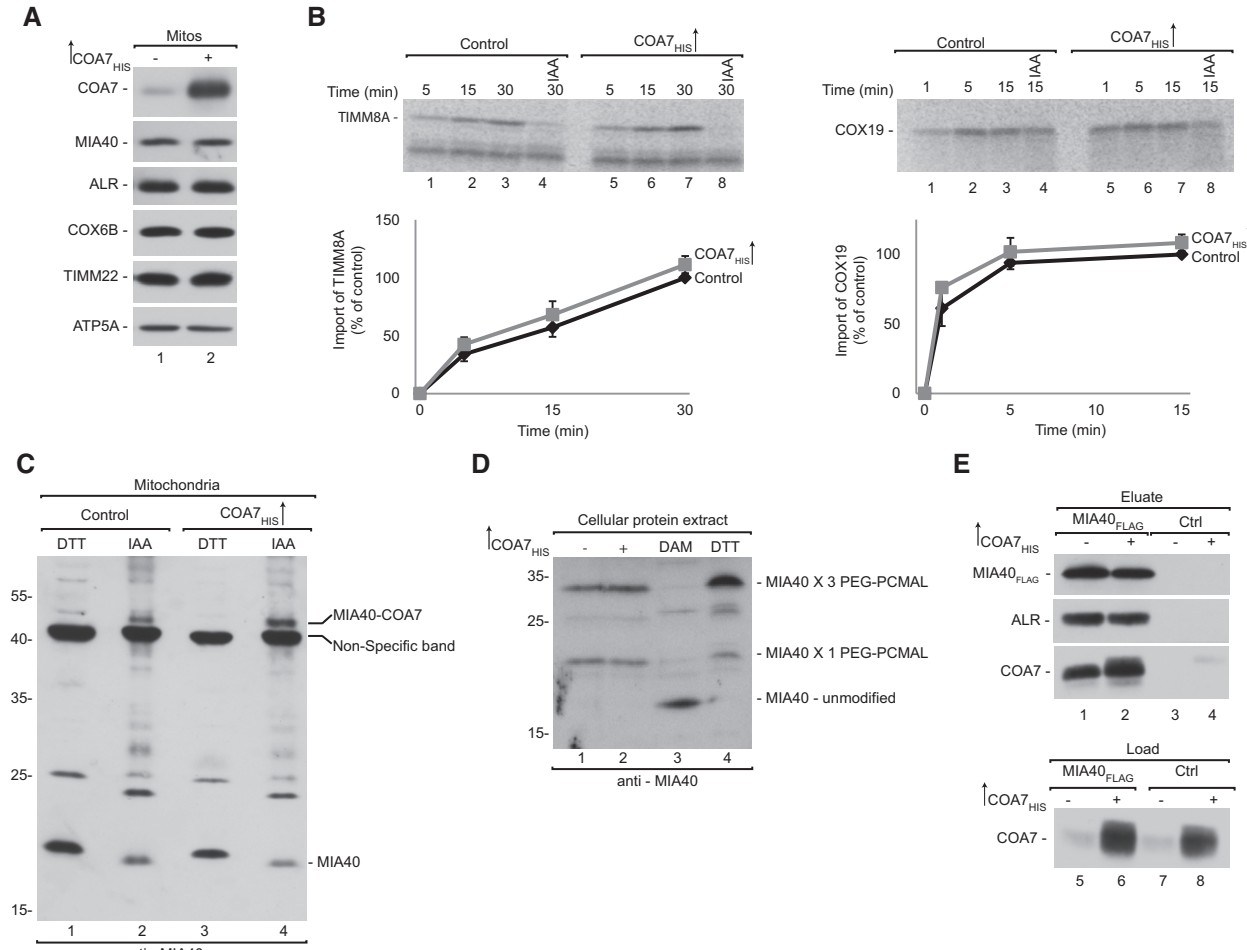

**Figure 3. COA7 does not influence the MIA pathway.**

A Mitochondria were isolated from cells that were transfected with a plasmid that encoded COA7$_{HIS}$ or an empty vector. Mitochondria were solubilized and analyzed by reducing SDS–PAGE and Western blot. Mitos, mitochondria.

B Radiolabeled [$^{35}$S]TIMM8A and [$^{35}$S]COX19 precursors were imported into mitochondria that were isolated from cells that were transfected with a plasmid that encoded COA7$_{HIS}$ or an empty vector. The samples were analyzed by reducing SDS–PAGE and autoradiography. The results of three biological replicates were analyzed, quantified, and normalized to control mitochondria at 30 min. The data are expressed as a mean ± SEM (n = 3). IAA, iodoacetamide.

C Mitochondria were isolated from cells that were transfected with a plasmid that encoded COA7$_{HIS}$ or an empty vector under reducing (DTT) and non-reducing (IAA) conditions and analyzed for levels of MIA40 by Western blot.

D Proteins from HEK293 cells transfected with empty plasmid or COA7$_{HIS}$ were modified with PEG-PCMal. Control cells were pretreated with the thiol-oxidizing agent diamide (DAM) or the reductant DTT. The samples were analyzed by SDS–PAGE and Western blot.

E Flp-In T-REx 293 cells induced to express MIA40$_{FLAG}$ were transfected with a plasmid that encoded COA7$_{HIS}$ or an empty vector. The affinity purification of MIA40$_{FLAG}$ was performed, and eluate fractions were analyzed by SDS–PAGE and Western blot.

Source data are available online for this figure.

MIA40 substrate TIMM8A was significantly augmented by MIA40$_{FLAG}$ overexpression (Fig 4B). Predictably, we also observed a significant increase in steady-state levels of COA7, similar to other classic MIA40-dependent proteins (e.g., COX6B and ALR; Fig 4C). Thus, we conclude that COA7 is imported by the MIA pathway and behaves like an MIA40 substrate. Considering the fact that ALR is an integral part of the MIA pathway, we also investigated whether ALR overexpression stimulates the import of COA7. Contrary to MIA40 overexpression, the import of COA7 was similar in ALR-overexpressed mitochondria compared with control mitochondria (Fig 4D). Likewise, ALR overexpression did not alter the

import of TIMM8A (Fig 4E). In agreement with the import results, steady-state levels of COA7 and MIA40-dependent proteins did not increase upon ALR overexpression (Fig 4F). We also examined the influence of MIA40 depletion on COA7 biogenesis. MIA40 knock-down using two different oligonucleotides decreased nearly 90% of MIA40 in HeLa cells (Fig 5A). As expected, the levels of COA7 were significantly reduced, similar to ALR and other substrates (e.g., COX6B and TIMM9). Proteins that are unrelated to the MIA pathway, such as actin, GAPDH, and ATP5A, were unchanged in MIA40 knockdown cells. This further substantiated the role of MIA40 in the biogenesis of COA7. Interestingly, ALR knockdown

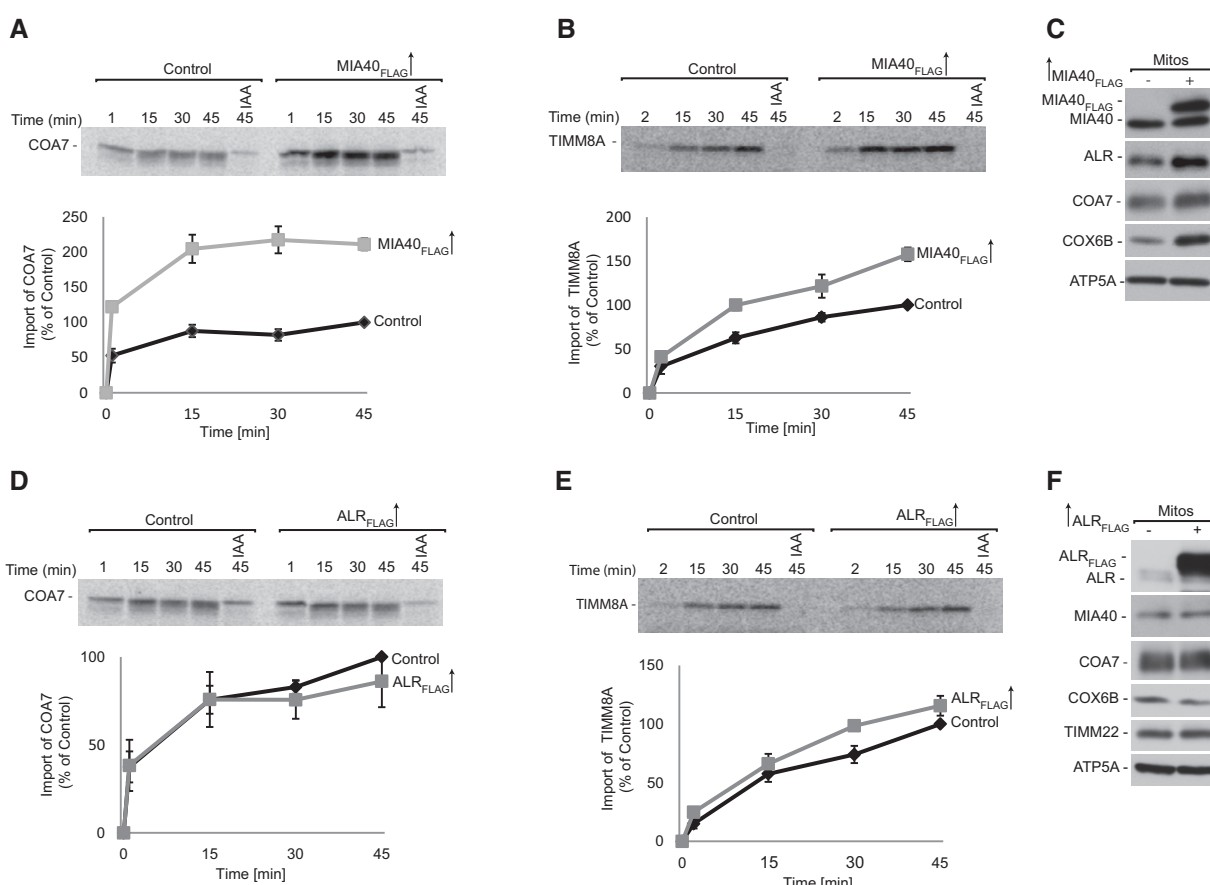

**Figure 4.  MIA40 facilitates COA7 import into mitochondria.**

A, B  Radiolabeled [$^{35}$S]COA7 (A) and [$^{35}$S]TIMM8A (B) precursors were imported into mitochondria that were isolated from Flp-In T-REx 293 cells induced to express MIA40$_{FLAG}$. The samples were analyzed by reducing SDS–PAGE and autoradiography. The results of three biological replicates were analyzed, quantified, and normalized to control mitochondria at 45 min. The data are expressed as a mean ± SEM ($n = 3$). IAA, iodoacetamide.

C  Mitochondria were isolated from Flp-In T-REx 293 cells induced to express MIA40$_{FLAG}$ and control cells. The samples were analyzed by SDS–PAGE and Western blot. Mitos, mitochondria.

D, E  Radiolabeled [$^{35}$S] COA7 (D) and [$^{35}$S] TIMM8A (E) precursors were imported into mitochondria that were isolated from Flp-In T-REx 293 cells induced to express ALR$_{FLAG}$ and control cells. The samples were analyzed by reducing SDS–PAGE and autoradiography. The results of three biological experiments were analyzed, quantified, and normalized to control mitochondria at 45 min. The data are expressed as a mean ± SEM ($n = 3$).

F  Mitochondria were isolated from Flp-In T-REx 293 cells induced to express ALR$_{FLAG}$ and control cells. The samples were analyzed by SDS–PAGE and Western blot.

Source data are available online for this figure.

also affected the steady-state levels of COA7 and MIA40-dependent proteins (Fig 5B).

To further substantiate the involvement of the MIA pathway in the import of COA7, we developed a cell line with deletion of the CPC motif in only one allele of the gene (HEK293 MIA40 WT/Del$^{53-60}$) (Fig 5C, Dataset EV2 and EV3). The Western blot analysis of HEK293 MIA40 WT/Del$^{53-60}$ cells revealed two bands that were specific to MIA40 (wild-type MIA40 and Del$^{53-60}$ MIA40; Fig 5D). In this cell line, the levels of wild-type MIA40 were partially affected by the presence of mutant allele suggesting a dominant negative effect of the mutation. In parallel, we observed lower levels of MIA40 substrates, including COX6B, TIMM9, and TIMM22. The levels of other mitochondrial proteins, such as MIC19 and ATP5A, remained unchanged (Fig 5D). The decrease in MIA40-dependent proteins was successfully rescued by the exogenous overexpression of MIA40 (Fig 5E, lanes 3 and 4). Thus, we established that the

observed phenotype was specific to MIA40 depletion. Furthermore, we used mitochondria that were isolated from MIA40 WT/Del$^{53-60}$ to measure the import of radiolabeled [$^{35}$S]COA7. The import efficiency of COA7 was reduced to almost 50% in these mutant cells when compared with wild-type cells (Fig 5F). We also observed a 40% reduction of the import of the classic MIA40 substrate TIMM8A (Fig 5G). Again, these observations indicate that COA7 utilizes the MIA40 pathway for import into mitochondria, and it is a newly identified non-canonical substrate of MIA40.

## COA7 pathological mutants are defective in mitochondrial import

Mutations in COA7 are associated with the defective assembly and function of respiratory chain complexes in patients diagnosed with mitochondrial encephalopathy (Martinez Lyons *et al*, 2016; Higuchi

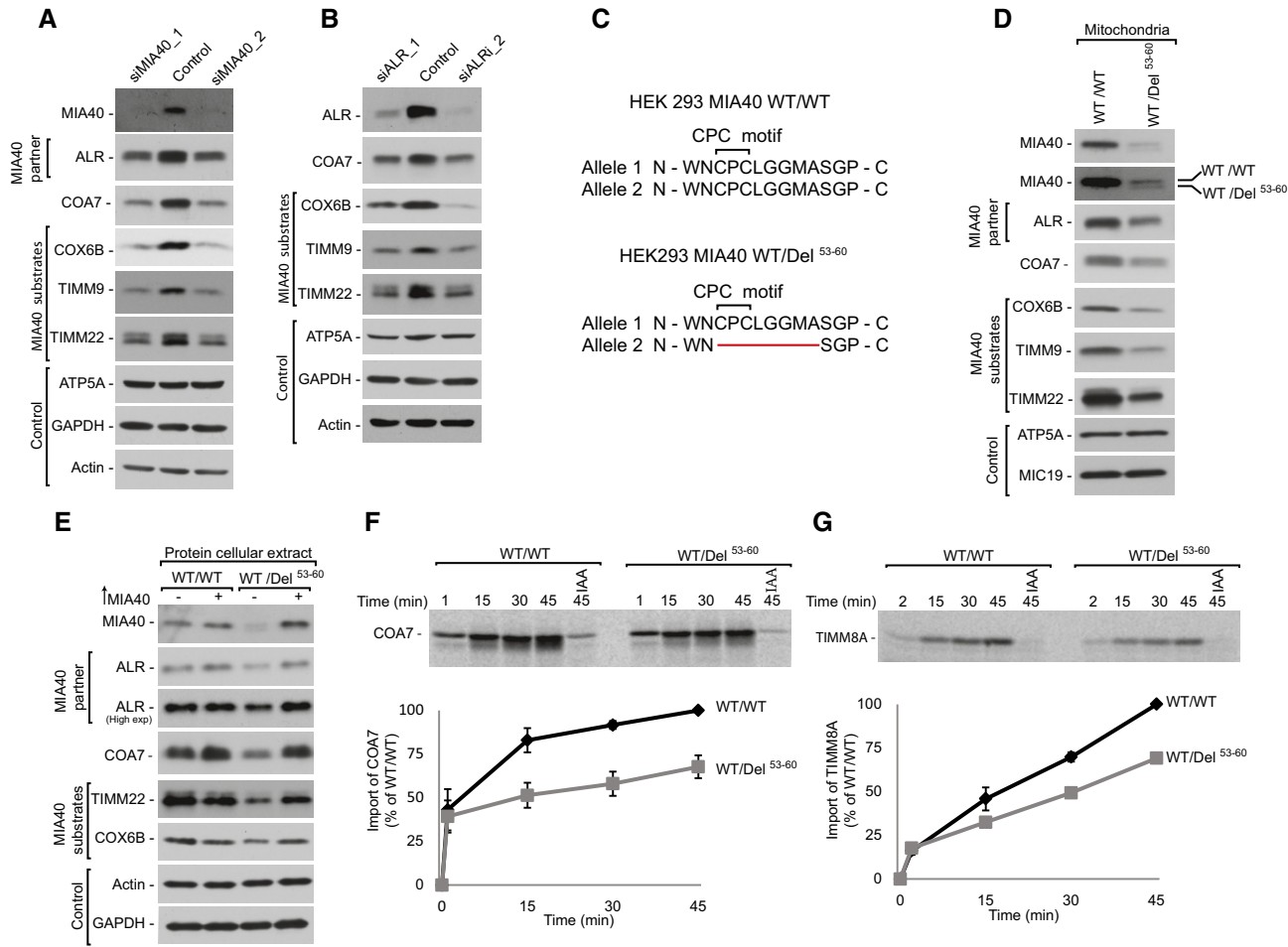

**Figure 5.  MIA40 is involved in the import and biogenesis of COA7.**

A   Cellular protein extracts were isolated from HeLa cells transfected with oligonucleotides that targeted different regions of MIA40 mRNA or with control oligonucleotides. The samples were analyzed by reducing SDS–PAGE and Western blot.

B   Cellular protein extracts were isolated from HeLa cells transfected with oligonucleotides that targeted different regions of ALR mRNA or with control oligonucleotides. The samples were analyzed by reducing SDS–PAGE and Western blot.

C   Schematic representation of MIA40 sequence containing the CPC motif in HEK293 MIA40 WT/WT and HEK293 MIA40 WT/Del[53-60] cells.

D   Cellular protein extracts were isolated from HEK293 MIA40 WT/WT and WT/Del[53-60] cells. The samples were analyzed by SDS–PAGE and Western blot.

E   Cellular protein extracts were isolated from HEK293 MIA40 WT/WT and WT/Del[53-60] cells transfected with a plasmid that encoded MIA40 or an empty vector. The samples were subjected to reducing SDS–PAGE and Western blot.

F, G   Radiolabeled [35S]COA7 (F) and [35S]TIMM8A (G) precursors were imported into mitochondria that were isolated from HEK293 MIA40 WT/WT and WT/Del[53-60] cells. The samples were analyzed by reducing SDS–PAGE and autoradiography. The results of three biological replicates were analyzed, quantified, and normalized to control mitochondria at 45 min. The data are expressed as a mean ± SEM ($n$ = 3). IAA, iodoacetamide.

Source data are available online for this figure.

---

et al, 2018). The first reported case presented with a biallelic compound heterozygous mutation, which led to two forms of COA7: COA7 with a single amino acid mutation (COA7-Y137C) and COA7 with a deletion of exon 2 (COA7-exon2$^\Delta$; Fig 6A). However, these mutant proteins were undetectable in patient-derived cultured skin fibroblasts (Martinez Lyons et al, 2016). Therefore, we characterized these COA7 mutants to elucidate the mechanism by which protein loss occurs under pathological conditions.

In order to do so, we transiently overexpressed wild-type COA7 and its mutant variants (COA7-Y137C$_{HIS}$ and COA7-exon2$^\Delta$$_{HIS}$) in HEK293 cells. We detected significantly lower levels of both variants compared with the wild-type (Fig 6B), which is consistent with

protein loss in fibroblasts from patients (Martinez Lyons et al, 2016). Moreover, we observed a decreased level of COA7-exon2$^\Delta$$_{HIS}$ than COA7-Y137C$_{HIS}$. Next, we investigated the localization of mutant proteins to the mitochondria isolated from these cells and observed a proportional decrease in the mitochondrial fraction together with a slight increase of the mutant proteins in the cytosolic fraction (Fig 6C, lanes 3, 9, and 12). Despite only a slight defect in mitochondrial localization of mutant COA7-Y137C$_{HIS}$, diminished steady-state levels of protein support the possibility of degradation. The mitochondrial fraction of COA7-Y137C$_{HIS}$ presents IMS localization, which was confirmed by mitoplasting (Appendix Fig S3A). As a control, we observed the efficient localization of mitochondrial

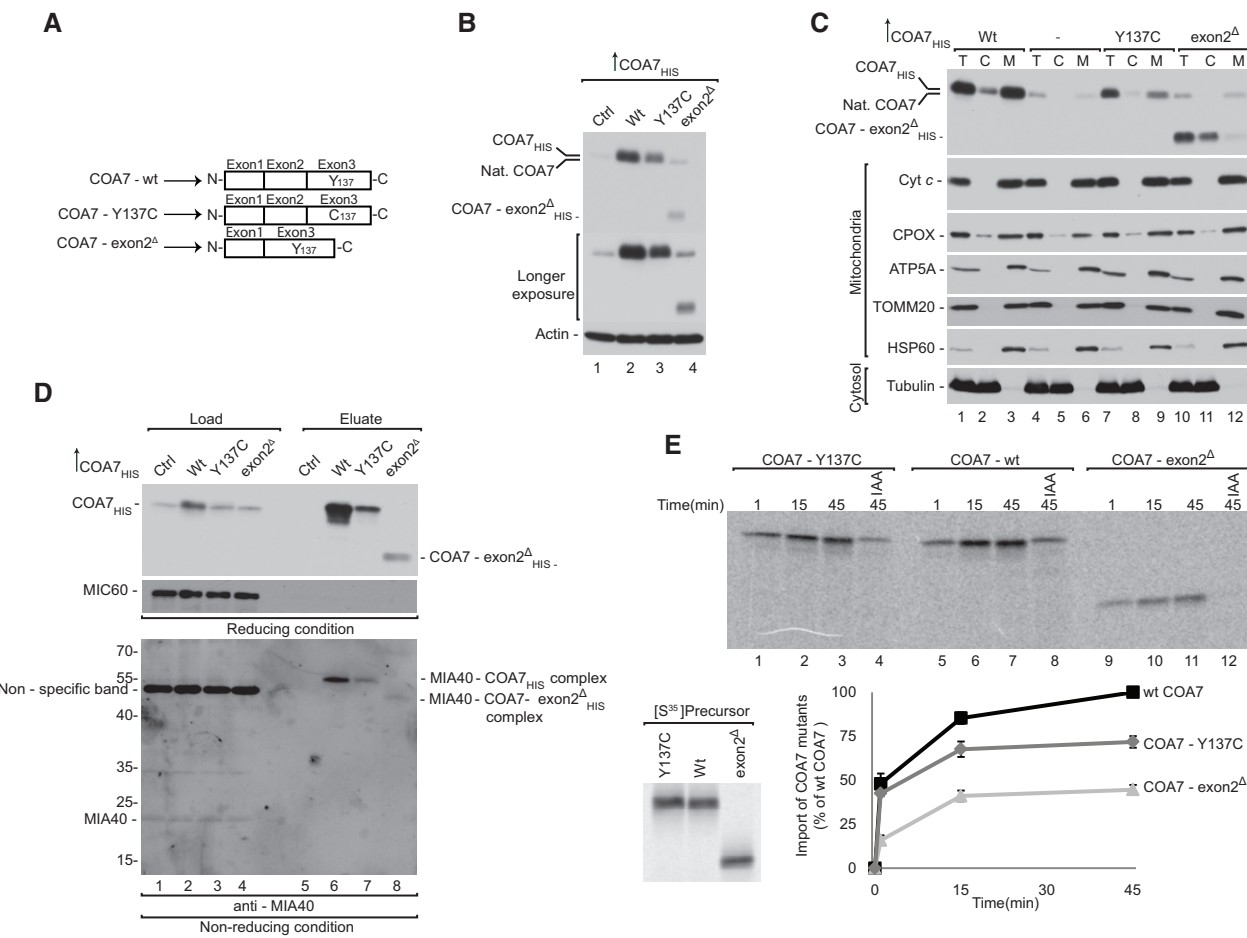

**Figure 6. COA7 pathological variants are import-defective.**

A  Schematic representation of wild-type and mutant COA7.
B  Cellular protein extracts were isolated from HEK293 cells that were transfected with a plasmid that encoded wild-type or mutant COA7. The samples were analyzed by reducing SDS–PAGE and Western blot.
C  Cellular fractions were prepared from HEK293 cells that were transfected with a plasmid that encoded wild-type or mutant COA7. The fractions were analyzed by reducing SDS–PAGE and Western blot. T, total; C, cytosol; M, mitochondria.
D  HEK293 cells that transiently expressed wild-type or mutant COA7$_{HIS}$ were solubilized, and the affinity purification of COA7$_{HIS}$ was performed. The samples were analyzed by reducing and non-reducing SDS–PAGE and Western blot. Load: 3%. Eluate: 100%.
E  Equal amounts of wild-type and mutant radiolabeled [$^{35}$S]COA7 precursors were imported into mitochondria isolated from HEK293 cells. The samples were analyzed by reducing SDS–PAGE and autoradiography. The results of three biological replicates were analyzed, quantified, and normalized to wild-type COA7 at 45 min. The data are expressed as a mean ± SEM (*n* = 3). IAA, iodoacetamide.

Source data are available online for this figure.

proteins, such as cytochrome *c*, ATP5A, TOMM20, and HSP60, in the mitochondrial fraction, whereas the cytosolic protein tubulin was present only in the cytosolic fraction.

Improperly folded mitochondrial IMS proteins can be substrates of proteases or can be retro-translocated to the cytosol (Bragoszewski *et al*, 2015). In order to understand the structural features that contribute to the instability of COA7-Y137C, we performed molecular modeling and simulation on the mutant protein. Homology modeling of the mutant suggested the possibility of two alternative disulfide bridges (C100-C137 and C111-C137) in addition to the native disulfide bond (C100-C111) that could potentially destabilize the structure (Appendix Fig S3B). To further understand the dynamics of the mutant proteins, we subjected the wild-type COA7 and

COA7-Y137C mutants to molecular dynamics simulations, which revealed greater flexibility in the two loop regions (Q159-K166 and D198-K202) of the COA7-Y137C (Appendix Fig S3C). We also identified the possibility of new salt bridge formation between lysine (K133) and aspartic acid (D136) in the COA7-Y137C mutant (Appendix Fig S3D). Thus, we propose that the additional cysteine at the 137[th] position leads to alternative disulfide bridges that compete with the native disulfide bond (C100-C111) causing misfolding and destabilization leading to structural instability, which can in turn increase the possibility of protein removal from the IMS. We previously established that COA7 interacts with MIA40 for its efficient import into mitochondria. Therefore, we investigated the influence of these mutations on the MIA40-COA7 interaction.

Affinity purification via COA7-Y137C$_{HIS}$ and COA7-exon2$^\Delta_{HIS}$ revealed an uncompromised interaction with MIA40 (Fig 6D, lanes 6–8). The lower level of co-purified MIA40 could be attributed to lower expression of the mutant variants (Fig 6D, lanes 2–4). Finally, the *in organello* import of the COA7 mutants revealed a 50% decrease in import efficiency for the COA7-Y137C mutant and a nearly 70% decrease for COA7-exon2$^\Delta$ compared with the wild-type (Fig 6E). Therefore, even though the mutant variants can interact with MIA40, their import into mitochondria is specifically impaired. As shown before, the steady-state levels of both mutated proteins are very low, which could be the result of a combination of their intrinsic instability, low import efficiency, and the inability of being productively maintained in the IMS, thus making them a subject of degradation.

### The proteasome degrades cytosol-localized COA7 variants

Mitochondrial IMS proteins are efficiently degraded by the proteasome either before import or after retro-translocation. We followed the degradation kinetics of mutant proteins in HEK293 cells transiently transfected with corresponding constructs by performing a cycloheximide-chase experiment (Fig 7A) and found that the mutant proteins were degraded faster than wild-type (Fig 7B, lanes 3 and 5, Appendix Fig S4A). We then evaluated the role of the proteasome in degradation of mutant proteins in these cells (Fig 7C). Effective inhibition of the proteasome by MG132 was confirmed by an increase in the ubiquitination of cellular proteins (Appendix Fig S4B). We found that MG132 treatment did not rescue the degradation of already synthesized mutant proteins (Fig 7D, lanes 2 and 3). In contrast, under active translation mutant COA7 was degraded by proteasome, while wild-type protein was only marginally affected, suggesting that proteins with slower rate of import to mitochondria were sensitive to proteasome-mediated degradation (Fig 7D, lanes 4 and 5, Appendix Fig S4C). Next, we investigated whether the stabilization of mutant proteins in the cytosol increases their translocation to mitochondria (Fig 7E). Indeed, in HEK293 transiently expressing COA7 in the presence of MG132-mediated proteasome inhibition, the levels of COA7 mutant proteins increased in the cytosol, which was paralleled by an increased mitochondrial content of the mutant proteins (Fig 7F, lanes 3, 7, 10, 14). Inhibition of the proteasome was confirmed by an increase in protein ubiquitination (Appendix Fig S4D). In order to further verify ubiquitin–proteasome system involvement in degradation of COA7, we co-expressed COA7$_{FLAG}$ with His-tagged ubiquitin and performed affinity purification via His-tag. We observed specific co-purification of COA7$_{FLAG}$ with markedly higher molecular weight characteristic for polyubiquitinated proteins (Fig EV3A). Mutant COA7 could also undergo degradation in the IMS by local proteases such as YME1L. To verify this hypothesis, we silenced YME1L in immortalized fibroblasts that were derived from the subject expressing pathogenic variants in COA7 (mt4229i) (Martinez Lyons et al, 2016). We observed only a moderate increase in both mutants of COA7 upon YME1L silencing (Fig EV3B). Of note, in the immortalized fibroblasts the lower levels of COA7-exon2$^\Delta$ in comparison with COA7-Y137C were observed. A similar phenomenon was noted in HEK293 under transient expression of COA7 variants, suggesting that the observed effect reflects specific propensities of these proteins and not a crosstalk between them.

### Rescue of COA7 mutant proteins in a patient cell line upon proteasome inhibition

We inhibited the proteasome machinery in mt4229i fibroblasts derived from patients with MG132 and other two clinically used proteasome inhibitors (bortezomib and carfilzomib). Proteasome inhibition increased the levels of both the mutation-carrying COA7 proteins (COA7-Y137C and COA7-exon2$^\Delta$) in the patient cells (Fig 8A). As a control for the experiment, we also observed an increase in the levels of HSP70 upon treatment with proteasome inhibitors as reported previously (Kim et al, 1999; Awasthi & Wagner, 2005). Interestingly, upon inhibition we also observed an increase in the levels of COX6A and COX6B, the structural subunits of complex IV which are affected in the patient. Further, we tested the localization of COA7 variants upon proteasome inhibition and observed an increase in their localization to mitochondria (Fig 8B). These results together with the previous findings using HEK293 cells substantiated the involvement of the proteasome in the degradation of misfolded/mislocalized mitochondrial proteins. Thus, we confirmed the influence of the proteasome system on protein levels under pathological conditions. Since proteasome inhibition increased COA7 levels inside mitochondria, we tested whether this could rescue the complex IV deficiency of patient cells (Martinez Lyons et al, 2016). Patient fibroblast presented 40% lower activity of complex IV than control fibroblasts (Fig 9A). Treatment of patient fibroblasts with proteasome inhibitors led to approximately 35% increase in complex IV activity (Fig 9B). To further verify whether the observed significant effect was attributable to higher levels of COA7, we combined bortezomib treatment with the overexpression of wild-type and the two different mutant versions of COA7. Fibroblasts that expressed wild-type COA7 and COA7-Y137C exhibited a significant 50% increase in complex IV activity, and COA7-exon2$^\Delta$ expression did not rescue the defective phenotype (Fig 9C). This strongly suggests that COA7-Y137C is a functionally competent protein that can effectively modulate the activity of complex IV if allowed to reach the IMS. Accordingly, neither of pathogenic mutants of COA7 had a negative effect on levels of complex IV subunits and on its activity when overexpressed in HEK293 cells in the presence of the native COA7 (Appendix Fig S5A and B). Thus, the defective phenotype that is observed in this patient is principally due to mislocalization and/or premature degradation of proteins by the proteasome system.

We then verified whether the rescue of complex IV activity by proteasome inhibition increased cell growth in oxidative phosphorylation-promoting galactose media. In the control fibroblasts, small concentrations of bortezomib (up to 2.5 nM) did not influence cell growth after 24 h in galactose media (Appendix Fig S5C). Interestingly, this was not the case for the patient fibroblasts that showed a tendency to an increase in growth (Appendix Fig S5D).

We further intended to analyze the levels of the respiratory chain complexes in the patient cells upon proteasome inhibition. Initially, when comparing patient cells with healthy control fibroblasts, the BN-PAGE analysis revealed a marked defect in the assembly of supercomplexes in patient cells observed with various antibodies (Fig 9D). In order to further confirm the role of COA7 in the observed increase in supercomplex formation, we overexpressed the wild-type COA7 and COA7-Y137C in patient

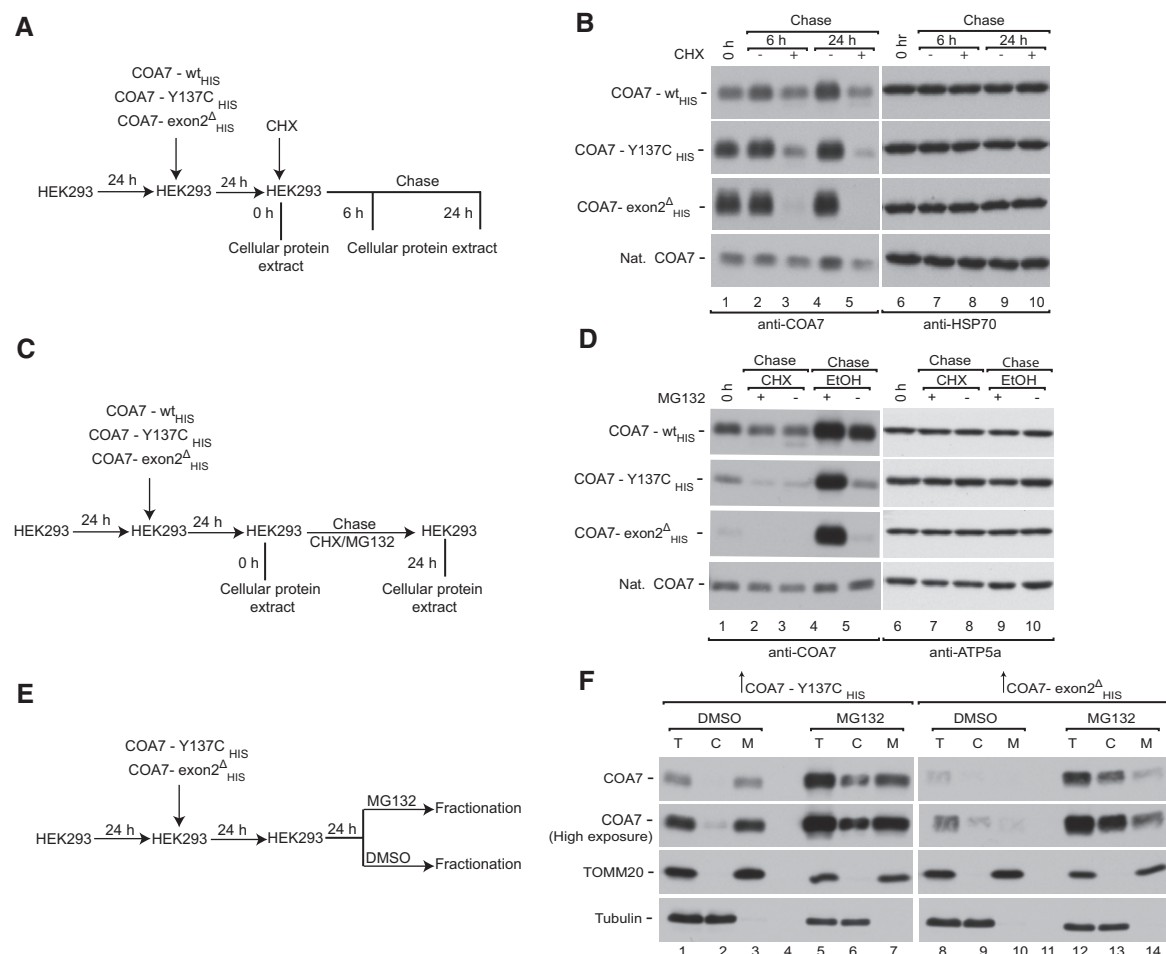

**Figure 7. The proteasome degrades cytosol-localized COA7 variants.**

A  Schematic representation of protein stability assay.

B  HEK293 cells that transiently expressed wild-type or mutant COA7_HIS were treated with CHX for the indicated times, and protein extracts were isolated. The samples were analyzed by reducing SDS–PAGE and Western blot.

C  Schematic representation of combined translation and proteasome inhibition assay.

D  HEK293 cells that transiently expressed wild-type or mutant COA7_HIS were treated with CHX and/or MG132 for the indicated times, and protein extracts were isolated. The samples were analyzed by reducing SDS–PAGE and Western blot. CHX, cycloheximide.

E  Schematic representation of the cellular localization assay.

F  Cellular fractions were prepared from HEK293 cells that transiently expressed mutant COA7_HIS and were treated with MG132. The samples were analyzed by reducing SDS–PAGE and Western blot.

Source data are available online for this figure.

fibroblasts. We analyzed cell extracts via BN-PAGE and confirmed an increased presence of supercomplexes (CI + CIII$_2$ and CI + CIII$_2$ + CIV) whenever wild-type COA7 or COA7-Y137C were overexpressed (Fig 9E).

We then verified whether the inhibition of proteasome could restore the formation of supercomplexes in the patient fibroblasts. Upon proteasome inhibition, we observed an increase in supercomplex formation using antibodies directed to complex I (NDUFS1), complex III (RIESKIE and UQCR1), and complex IV (COX4 and COX5B) (Fig 9F). Thus, we propose that the degradation of slow-import mutant COA7 can be prevented by proteasome inhibition leading to partial restoration of COA7 levels in mitochondria and of respiratory chain function (Fig 10).

# Discussion

The present study established a recently identified IMS protein, COA7, as a non-canonical substrate of MIA40. COA7 exists as an oxidized protein in the IMS of mitochondria, and MIA40 facilitates its import into IMS (Fig 10). Whole-exon sequencing of a patient who was diagnosed with leukoencephalopathy recently revealed biallelic heterozygous mutations of COA7 that led to absence of the protein (Martinez Lyons et al, 2016). We discovered that the proteasome-mediated degradation of pathogenic COA7 mutant proteins accounted for nearly the complete absence of the protein. When cells expressing both point mutation and deletion instable COA7 proteins were treated with proteasome inhibitors, the degradation of

**A**

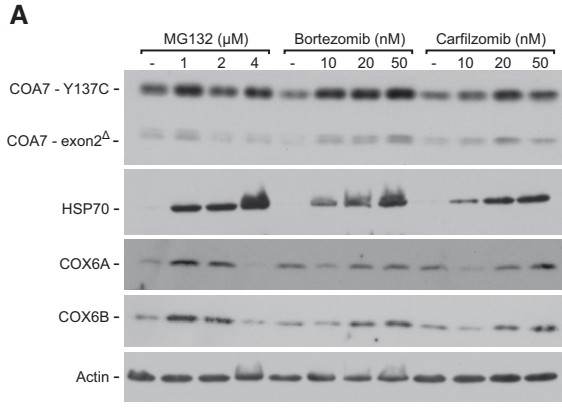

**B**

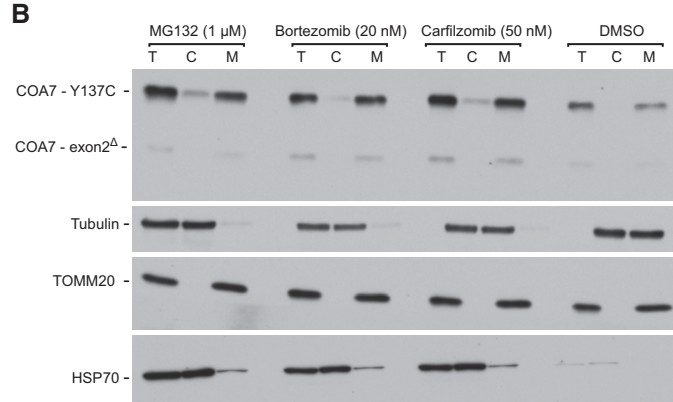

**Figure 8.  Inhibition of the proteasome rescues the mitochondrial levels of COA7 in patient-derived fibroblasts.**

A   Cellular protein extracts were isolated from immortalized patient-derived skin fibroblasts that were treated with the indicated concentrations of MG132, bortezomib, or carfilzomib. The samples were analyzed by reducing SDS–PAGE and Western blot.

B   Cellular fractions were prepared from immortalized patient-derived skin fibroblasts that were treated with the indicated concentrations of MG132, bortezomib, or carfilzomib. The samples were analyzed by reducing SDS–PAGE and Western blot. T, total; C, cytosol; M, mitochondria.

Source data are available online for this figure.

the mutated proteins was prevented and their localization to mitochondria was increased. In the patient-derived cells, this increase in COA7 levels was associated with an increase in complex IV activity and improved formation of the respiratory chain supercomplexes (Fig 10).

Most mitochondrial proteins are translated in the cytosol where they are inevitably in proximity to the proteasome, which is the main cytosolic protein quality control machinery. It is known that proteasome is involved in degradation of several proteins from the outer mitochondrial membrane and the IMS (Pearce & Sherman, 1997; Zhong et al, 2005; Azad et al, 2006; Yonashiro et al, 2006; Karbowski et al, 2007; Neutzner et al, 2008; Radke et al, 2008; Ziviani et al, 2010; Bragoszewski et al, 2015). Mitochondrial proteins undergo proteasomal degradation as a turnover mechanism of mature proteins but also as an effect of mislocalization during mitochondrial protein biogenesis. Upon the failure of mitochondrial protein import, excess mitochondrial precursor proteins in the cytosol are eliminated by the proteasome-mediated degradation prior to their import (Bragoszewski et al, 2013; Wrobel et al, 2015). Moreover, substrates of the MIA pathway in yeast were found to be ubiquitinated and accumulated in response to proteasome inhibition, even in the presence of active import machinery (Bragoszewski et al, 2013; Kowalski et al, 2018). Thus, the proteasome system acts as a vital checkpoint for the improper localization of proteins and promotes efficient mitochondrial IMS protein biogenesis (Bragoszewski et al, 2013, 2017). The present data demonstrate that mislocalized COA7 is more prone to degradation. Interestingly, interaction of mutant variants of COA7 with MIA40 raises the possibility that they follow retro-translocate-mediated degradation due to inefficient folding and accumulation in mitochondria (Bragoszewski et al, 2015).

In the present study, we characterized two pathogenic mutants of COA7 that are associated with mitochondrial leukoencephalopathy. These proteins were less efficiently imported into mitochondria, which led to proteasomal degradation in the cytosol.

Inhibition of the proteasome rescued mutant protein levels and increased their mitochondrial levels, accompanied by higher levels of complex IV subunits (e.g., COX6B and COX6A) and an increase in the activity of complex IV. The recovery of complex IV activity was also achieved by the overexpression of COA7-Y137C protein, suggesting that the mutant protein, if allowed to be imported into mitochondria, was sufficiently active to ameliorate the pathological phenotype. Analogously, other studies reported that the overexpression of mutant versions of mitochondrial proteins, such as NUBPL and FOXRED1, was able to reverse the pathological phenotype in patient cells (Tucker et al, 2012; Formosa et al, 2015). The stimulation of mitochondrial import of the pathogenic mutant CHCHD10 was recently proposed as a possible therapeutic strategy for amyotrophic lateral sclerosis (Lehmer et al, 2018). Our data suggest that the inhibition of excessive degradation by the proteasome can rescue mitochondrial function in diseases that are associated with the mislocalization and premature degradation of mitochondrial proteins.

Pathogenic mutations of various mitochondrial proteins (e.g., NDUFAF3, NDUFAF4, FOXRED1, COA5, COA6, COX6B, CHCHD10, and tafazzin) were associated with the secondary loss of other mitochondrial proteins, which can aggravate the disease (Nouws et al, 2012; Dudek et al, 2013; Modjtahedi & Kroemer, 2016). In patient-derived fibroblasts, structural subunits of cytochrome c oxidase COX6A and COX6B decreased in parallel with COA7 and were rescued by proteasome inhibition. Thus, proteasome inhibition not only rescued COA7 levels but also prevented the secondary loss of mitochondrial proteins. Notably, mitochondrial biogenesis has been previously shown to be a rescue mechanism for defective mitochondria (Hansson et al, 2004; Kuhl et al, 2017).

Mitochondrial diseases affect approximately one in 2,000 people and may arise at any age with a wide range of clinical symptoms (Gorman et al, 2016; Suomalainen & Battersby, 2018). Among these diseases are several genetic mitochondrial pathologies that are associated with the apparent loss of mutated proteins that are often

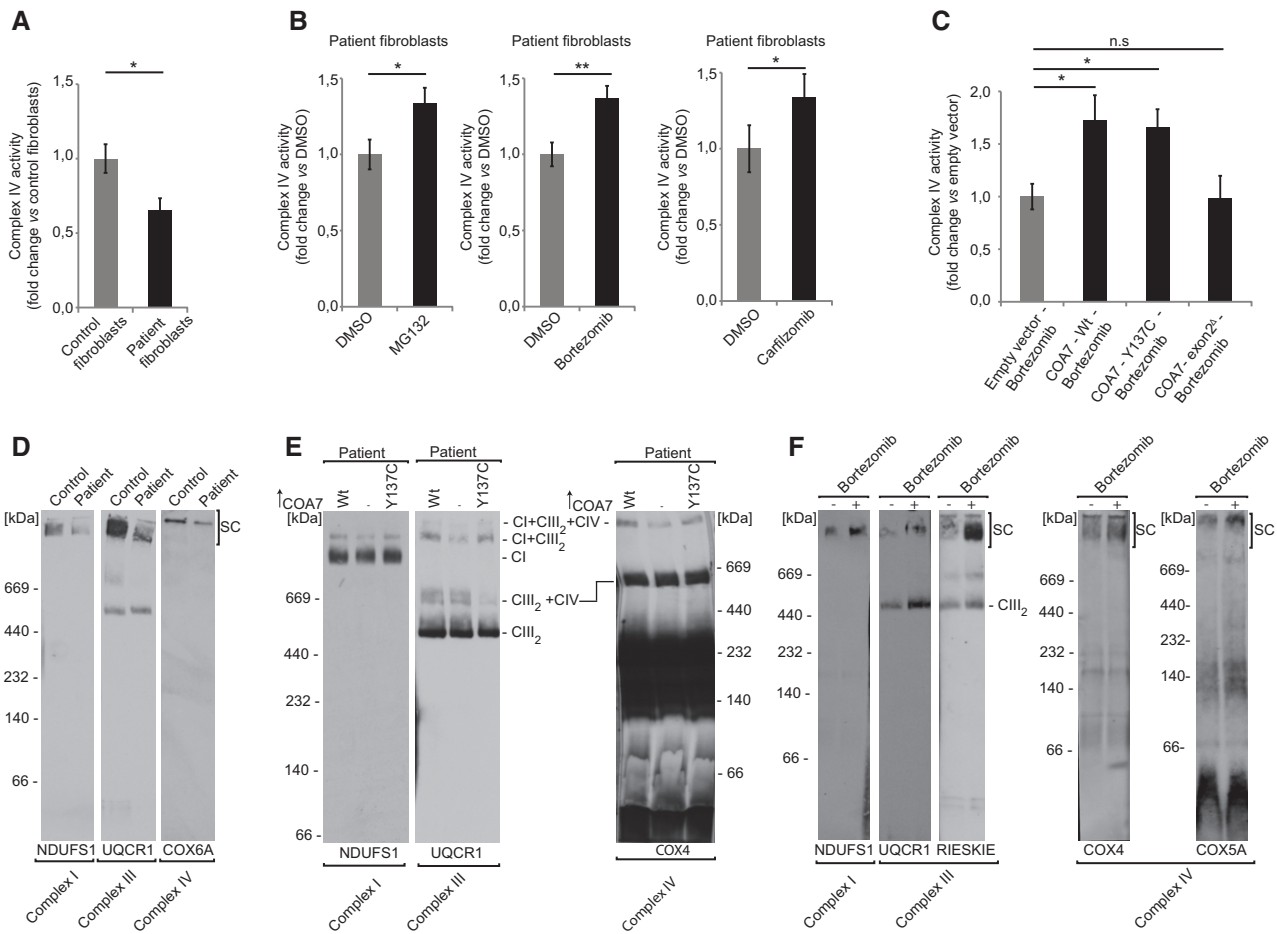

**Figure 9.  Inhibition of the proteasome rescues the activity and assembly of the respiratory complexes.**

A   Complex IV activity was assessed by oxidation of cytochrome *c* in digitonized cellular extracts from the immortalized patient-derived skin fibroblasts and control healthy fibroblasts. The results of three biological replicates were quantified and normalized to control fibroblasts. Data are expressed as a mean ± SEM (*n* = 3, *$P$ = 0.01) (two-tailed Student's *t*-test).

B   Complex IV activity was assessed by oxidation of cytochrome *c* in digitonized cellular extracts from immortalized patient-derived skin fibroblasts treated with MG132 (1 μM), bortezomib (20 nM), or carfilzomib (50 nM) for 12 h and recovered for another 6 h. The results of three biological replicates were quantified and normalized to DMSO-treated samples, and the data are expressed as a mean ± SEM (*n* = 3, *$P$ = 0.04 [DMSO vs MG132], **$P$ = 0.0025 [DMSO vs Bortezomib], *$P$ = 0.02 [DMSO vs Carfilzomib]) (two-tailed Student's *t*-test).

C   Immortalized patient-derived skin fibroblasts transiently expressing wild-type or mutant COA7 (COA7-Y137C and COA7-Ex2$^\Delta$) for 48 h were treated with bortezomib (20 nM) during the last 12 h. The fibroblasts were harvested, and complex IV activity was measured in digitonized cellular extracts. The results of three biological replicates were quantified and normalized to empty vector bortezomib-treated samples, and the data are expressed as a mean ± SEM (*n* = 3, *$P$ = 0.01 [Empty vector vs. COA7- Wt], *$P$ = 0.005 [Empty vector vs COA7 – Y137C]) (two-tailed Student's *t*-test).

D   Mitochondria isolated from control and patient fibroblast were solubilized in digitonin buffer and analyzed by 4–13% gel BN-PAGE and Western blot. SC, supercomplexes.

E   Immortalized patient-derived skin fibroblasts transfected with plasmid encoding wild-type COA7$_{HIS}$ or COA7-Y137C$_{HIS}$ were solubilized in DDM buffer and analyzed by 4–13% gel BN-PAGE and Western blot.

F   Mitochondria were isolated from immortalized patient-derived skin fibroblasts treated with bortezomib (10 nM) for 12 h and recovered for another 6 h. Mitochondria were solubilized in digitonin buffer and analyzed by 4–13% gel BN-PAGE and Western blot. SC, supercomplexes.

associated with the more generalized deficiency of mitochondrial protein biogenesis (Modjtahedi & Kroemer, 2016). Despite dramatic improvements in the genetic and metabolic diagnoses of these severe progressive diseases, no curative treatments have been discovered. The current treatment regimen for respiratory chain deficiency is mostly metabolite supplementation with CoQ10, creatinine, or riboflavin, which is only a symptomatic treatment and does not ameliorate the underlying pathological condition. In the

present study, we used bortezomib and carfilzomib (i.e., two proteasome inhibitors that have been approved for the treatment of patients with multiple myeloma and mantle cell lymphoma) to reverse the pathological phenotype in fibroblasts that were derived from a patient who suffered from mitochondrial leukoencephalopathy. The possibility of using clinically approved inhibitors, such as bortezomib, against this type of disease requires careful consideration due to possible side effects (Cavaletti *et al*, 2007; Nowis *et al*,

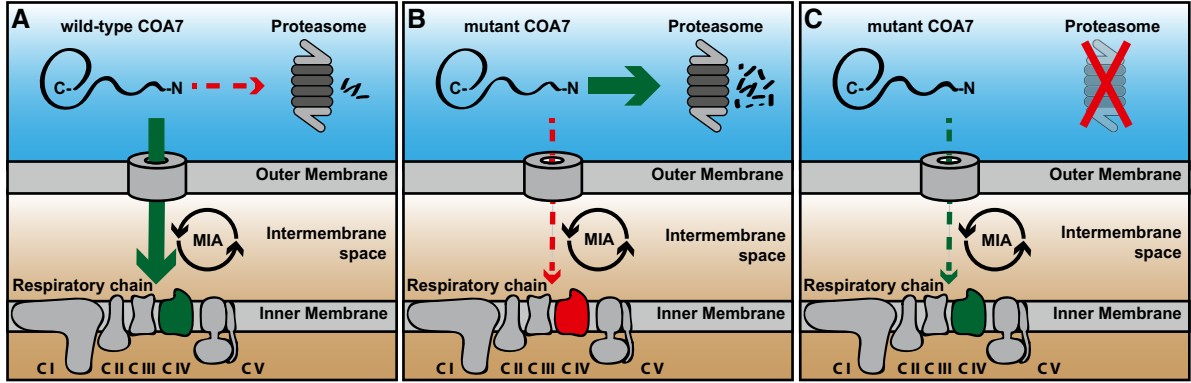

**Figure 10. The role of MIA pathway and proteasome in the biogenesis of COA7.**

A    The precursor form of wild-type COA7 is synthesized in the cytosol and imported into the mitochondrial intermembrane space by the MIA pathway, where it is involved in the respiratory chain biogenesis.

B    The precursor form of mutant COA7 is import-defective, and therefore, it is degraded in the cytosol by proteasome.

C    Proteasome inhibition rescues the mitochondrial import of mutant COA7 as well as the biogenesis and function of the respiratory chain.

2010; Zheng *et al*, 2012; Kaplan *et al*, 2017). Yet, the severity of mitochondrial diseases prompts us to suggest that proteasome inhibitors may be applied in the course of a therapeutic intervention to treat mitochondrial diseases that are associated with the reduced steady-state levels and excessive degradation of mitochondrial proteins by the proteasome that are primarily attributable to the pathogenic mutations (Fig 10).

## Materials and Methods

### Cell lines and growth conditions

HeLa, HEK293, and Flp-In T-REx 293 cells were cultured at 37°C with 5% $CO_2$ in standard Dulbecco's modified Eagle's medium (DMEM) supplemented with 10% (v/v) heat-inactivated fetal bovine serum, 2 mM L-glutamine, 100 U/ml penicillin, and 10 μg/ml streptomycin sulfate. Flp-In T-REx 293 with inducible expression of MIA40$_{FLAG}$, MIA40-C53S$_{FLAG}$, MIA40-C55S$_{FLAG}$, and MIA40-C53,55S$_{FLAG}$ (SPS), and ALR$_{FLAG}$ were generated by co-transfection with pcDNA5/FRT/TO plasmid containing respective cDNA and a helper plasmid pOG44. Single cell-derived cell lines were obtained by selection with hygromycin. Expression of respective proteins was induced by treatment with tetracycline (100 ng/ml) for 24 h. Cell lines with introduced MIA40-C55S$_{FLAG}$ and SPS showed reproducibly lower levels of expression than MIA40$_{FLAG}$ and MIA40-C53S$_{FLAG}$. Immortalized skin fibroblast cell lines (Ai and mt4229i) were grown in standard DMEM supplemented with 10% (v/v) heat-inactivated fetal bovine serum, 2 mM L-glutamine, 100 U/ml penicillin, 10 μg/ml streptomycin sulfate, 1 mM sodium pyruvate, and 50 μg/ml uridine. The MIA40 deletion cell line (HEK293 MIA40 WT/Del Del[53-60]) was generated using guide RNA that targeted exon 3 of MIA40 ORF cloned into the pX330 vector (Addgene, 42230). The recombinant plasmid was co-transfected into HEK293 cells with green fluorescent protein (GFP)-expressing plasmid. Single cells that expressed GFP were sorted into each well of the 96-well plate and grown at 37°C under 5% $CO_2$. Deletion that corresponded to amino

acids 53–60 of MIA40 was confirmed by sequencing and CRISPR ID analysis. Where indicated, cells were grown in DMEM that contained low glucose (1.1 g/l) or galactose (1.8 g/l) as described below.

### Human samples

Immortalized skin fibroblast cell lines (Ai and mt4229i) reported in Martinez Lyons *et al* (2016), were obtained from the BioBank of Telethon Italy located at the Fondazione Istituto Neurologico Carlo Besta, Milan, Italy. Informed consent was obtained from all subjects. All the procedures involving human samples conformed to the principles set out in the WMA Declaration of Helsinki and the Department of Health and Human Services Belmont Report.

### DNA and siRNA transfection

For protein overexpression, HEK293 cells were seeded at $0.07 \times 10^6/cm^3$ and grown in low-glucose DMEM for 24 h. Medium was exchanged to galactose DMEM, and cells were transfected with $0.05 \ \mu g/cm^3$ pcDNA3.1(+) containing the gene of interest mixed with $0.1 \ \mu l/cm^3$ Gene Juice Transfection Reagent (70967; Merck Millipore) according to the manufacturer's protocol. Cells were incubated at 37°C and 5% $CO_2$ for 24 h. Cells transfected with empty pcDNA3.1 (+) were used as a control.

For gene knockdown, the following mRNA sequences were targeted: MIA40_1 (5′-ATAGCACGGAGGAGATCAA-3′), MIA40_2 (5′-GGAATGCATGCAGAAATAC-3′), ALR_1 (5′-GGAGTGTGCTGA AGACCTA-3′), ALR_2 (5′-GCATGCTTCACACAGTGGCTGT-3′), COA 7_1 (5′-GATGGTGTTGATAAGGATGA-3′), and COA7_2 (5′-GGCAC ATGATGGACAGGTTAA-3′), YME1L (5′-GATGCATTTAAAACTG GTTTT-3′). The transfection of HEK293 cells was performed in low-glucose DMEM using oligofectamine as per the manufacturer's instructions. After 24 h in low glucose, the medium was changed to galactose-containing DMEM for the next 48 h and cells were harvested. As a control, the cells were transfected with Mission siRNA universal negative control (SIC001, Sigma).

## Antibodies

The antibodies against following proteins were used in the study: actin (Sigma, A1978, 1:500), ALR (Santa Cruz Biotechnology, Sc-134869, 1:500), ATP5A (Abcam, ab14748, 1:500), COA7 (Sigma, HPA029926, 1:500), COX4 (Abcam, ab14744, 1:500, and Cell Signaling Technology, 4850, 1:2,000), COX5B (Santa Cruz Biotechnology, Sc-374417, 1:500), COX6A (Rabbit Serum, 3282.7, 1:1,000), COX6B (Abcam, ab110266, 1:500), COX17 (Proteintech, 11464-1-AP, 1:100), GAPDH (Santa Cruz Biotechnology, Sc-47724, 1:1,000), HSP60 (Sigma, H4149, 1:500), MIA40 (Rabbit Serum, WA136-5, 1:500), MIC60 (Novus Biologicals, NB100-1919, 1:1,000), NDUFS1 (Santa Cruz Biotechnology, Sc-50132 1:1,000), SDHB (Santa Cruz Biotechnology, Sc-25851, 1:500), TIMM9 (Abcam, ab57089, 1:200), TIMM22 (Proteintech, 14927-1-AP, 1:500), TOMM20 (Santa Cruz Biotechnology, Sc-11415, 1:500), SDHA (Santa Cruz Biotechnology, Sc-166947, 1:1,000), YME1L (Proteintech, 11510-1-AP, 1:100), ubiquitin (Santa Cruz Biotechnology, Sc-8017, 1:500), UQCR1 (Sigma, HPA002815, 1:500).

## Cellular protein lysate

Cells were harvested and lysed in radioimmunoprecipitation assay (RIPA) buffer (65 mM Tris–HCl [pH 7.4], 150 mM NaCl, 1% v/v NP 40, 0.25% sodium deoxycholate, 1 mM ethylenediaminetetraacetic acid [EDTA], and 2 mM phenylmethylsulfonyl fluoride [PMSF]) for 30 min at 4°C. The lysate was clarified by centrifugation at 14,000 × $g$ for 30 min at 4°C. The supernatant was collected, and the protein concentration was measured by the bicinchoninic acid protein assay (Thermo Fisher Scientific). The supernatant was diluted in Laemmli buffer that contained either 50 mM DTT or 50 mM IAA for reducing and non-reducing conditions, respectively.

## Mitochondrial isolation and fractionation

HEK293 cells were harvested and resuspended in ice-cold trehalose buffer (10 mM HEPES-KOH [pH 7.7], 300 mM trehalose, 10 mM KCl, 1 mM EDTA, and 1 mM ethylene glycol-bis[β-aminoethyl ether]-N,N,N',N'-tetraacetic acid [EGTA]) supplemented with 2 mg/ml bovine serum albumin (BSA) and 2 mM PMSF. After homogenization in a Dounce glass homogenizer (Sartorius), the homogenate was clarified by centrifugation at 1,000 × $g$ for 10 min at 4°C. The supernatant was centrifuged at 10,000 × $g$ for 10 min at 4°C. The pellet was resuspended in trehalose buffer without BSA, and protein levels were quantified using the Bradford assay.

Human fibroblasts were harvested and resuspended in ice-cold isotonic buffer (10 mM MOPS [pH 7.2], 75 mM mannitol, 225 mM sucrose, and 1 mM EGTA) supplemented with 2 mg/ml BSA and 2 mM PMSF, and subjected to centrifugation at 1,000 × $g$ for 5 min at 4°C. The cell pellet was then resuspended in cold hypotonic buffer (10 mM MOPS [pH 7.2], 100 mM sucrose, and 1 mM EGTA) and incubated on ice for 5–7 min. The cell suspension was homogenized in a Dounce glass homogenizer (Sartorius). Cold hypertonic buffer (1.25 M sucrose and 10 mM MOPS [pH 7.2]) was added to the cell homogenate (1.1 ml/g of cells). The homogenate was subjected to centrifugation at 1,000 × $g$ for 10 min at 4°C to pellet the cellular debris. The supernatant that contained mitochondria was then carefully aspirated and centrifuged again. The supernatant

was then subjected to high-speed centrifugation at 10,000 × $g$ for 10 min at 4°C to pellet mitochondria. The pellet was resuspended in isotonic buffer without BSA and quantified using the Bradford assay. Mitochondria were denatured in 2× Laemmli buffer with 50 mM DTT or 50 mM IAA. The samples were separated by SDS–PAGE and analyzed by Western blot.

For cell fractionation, after homogenization, the supernatant was then divided into three equal aliquots. One aliquot was saved as the total fraction. Two aliquots were centrifuged at 10,000 × $g$ for 10 min at 4°C to collect the pellet and supernatant for the mitochondrial and cytosolic fractions, respectively. The proteins from the total and cytosolic fractions were precipitated by pyrogallol red. Finally, all three fractions were denatured in urea sample buffer with 50 mM DTT for SDS–PAGE and Western blot.

## Mitoplasting

Isolated mitochondria were incubated on ice for 30 min in either sucrose buffer (250 mM sucrose and 20 mM HEPES/KOH [pH 7.4]) or swelling buffer (25 mM sucrose and 20 mM HEPES/KOH [pH 7.4]). The solution was divided into two equal parts. One part was treated with proteinase K (25 μg/ml) for 5 min on ice, and another part was left untreated. Proteinase K was inactivated by the addition of 2 mM PMSF. Mitochondria were pelleted by centrifugation, and the supernatant fraction was subjected to pyrogallol red precipitation to recover the proteins. The pellet was then denatured in urea sample buffer with 50 mM DTT and analyzed by SDS–PAGE and Western blot.

## FLAG-tag affinity purification

A total of $4 \times 10^6$ Flp-In T-REx 293 derived cells were seeded and grown in low-glucose medium for 24 h and then induced with tetracycline (100 ng/ml) in galactose medium for 24 h. The cells were harvested and solubilized in lysis buffer (50 mM Tris–HCl [pH 7.4], 150 mM NaCl, 10% glycerol, 1 mM EDTA, and 1% digitonin) supplemented with 2 mM PMSF and 50 mM IAA for 20 min at 4°C. The lysate was clarified by centrifugation at 20,000 $g$ for 15 min, and the supernatant was incubated with anti-FLAG M2 affinity gel (Sigma) for 2 h at 4°C with mild rotation. After binding, the resin was washed three times with lysis buffer without digitonin. The column-bound proteins were eluted with either 3X FLAG peptide for mass spectrometry or Laemmli buffer under reducing or non-reducing conditions for SDS–PAGE and Western blot.

## COA7$_{HIS}$ affinity purification

HEK293 cells were grown in low-glucose DMEM for 24 h. The cells were then transferred to DMEM and transfected with pcDNA3.1(+)-COA7$_{HIS}$ using Gene Juice Transfection Reagent (Merck Millipore) according to the manufacturer's protocol. The cells were harvested 24 h after transfection and solubilized in ice-cold digitonin buffer (1% digitonin, 10% glycerol, 20 mM Tris–HCl [pH 7.4], 100 mM NaCl, and 20 mM imidazole [pH 7.4]) supplemented with 2 mM PMSF and 50 mM IAA for 20 min at 4°C. The lysate was clarified by centrifugation at 20,000 $g$ for 15 min at 4°C. The supernatant was then incubated with Ni-NTA agarose beads (Invitrogen) with mild rotation for 1.5 h at 4°C. After binding, the beads were washed three

times with wash buffer (20 mM Tris–HCl [pH 7.4], 100 mM NaCl, and 35 mM imidazole [pH 7.4]). Bead-bound proteins were eluted with Laemmli buffer that contained either 50 mM DTT or 50 mM IAA for reducing and non-reducing conditions, respectively. The samples were denatured and subjected to SDS–PAGE and Western blot.

### Ubiquitin$_{HIS}$ affinity purification

HEK293 cells were grown in low glucose for 24 h and co-transfected with plasmids encoding ubiquitin$_{HIS}$ and COA7$_{FLAG}$ and grown in galactose medium for 24 h in the presence of 1 mM MG132. The cells were harvested and resuspended in denaturing lysis buffer (6 M guanidine hydrochloride, 100 mM potassium phosphate buffer [pH 8.0], 10 mM Tris–HCl [pH 8.0], 50 mM iodoacetamide, 20 mM imidazole, 0.1% Triton X-100, 2 mM PMSF, and 1 mM MG132) for 15 min at room temperature. The cell debris was pelleted by centrifugation for 15 min at 14,000 *g* at room temperature. The load fractions were collected from the supernatant fraction. The remaining supernatant was then incubated with Ni-NTA agarose beads (Qiagen). The samples were incubated with Ni-NTA for 2 h at room temperature with gentle mixing. After 2 h, the suspension was subjected to centrifugation for 2 min at 200 × *g* and the supernatant was discarded. Ni-NTA beads were washed once with lysis buffer and three times with wash buffer (8 M urea, 100 mM KPi [pH 6.4], and 10 mM Tris–HCl [pH 6.4], 30 mM imidazole). The protein bound to Ni-NTA beads was eluted by resuspending in 2× Laemmli buffer (4% SDS, 20% glycerol, 125 mM Tris–HCl [pH 6.8], and 0.02% bromophenol blue) containing 100 mM DTT at 65°C with vigorous shaking. Load fraction was precipitated with 10% trichloroacetic acid (TCA), washed with acetone, air-dried, and resuspended with 2× Laemmli buffer supplemented with 100 mM DTT. All of the samples were denatured at 65°C and subjected to SDS–PAGE and Western blotting.

### Radioactive precursor synthesis and *in organello* import

The cDNA of precursors was cloned into a pTNT vector under the SP6 promoter, and radiolabeled precursors were synthesized using [$^{35}$S]methionine and the TNT SP6 Quick Coupled Transcription/ Translation system (Promega). The precursors were precipitated by ammonium sulfate and reduced in urea buffer (6 M urea and 60 mM MOPS-KOH [pH 7.2]) with 50 mM DTT. The import of radiolabeled precursors into isolated mitochondria was performed at 30°C in import buffer (250 mM sucrose, 80 mM potassium acetate, 5 mM magnesium acetate, 5 mM methionine, 10 mM sodium succinate, 5 mM adenosine triphosphate, and 20 mM HEPES/KOH [pH 7.4]). Import was stopped by the addition of 50 mM IAA and incubation on ice. Non-imported precursors were removed by proteinase K treatment for 15 min. Proteinase K was then inactivated by the addition of 2 mM PMSF, and mitochondria were washed with high-sucrose buffer (500 mM sucrose and 20 mM HEPES/KOH [pH 7.4]). The samples were solubilized in Laemmli sample buffer with 50 mM DTT and 0.2 mM PMSF and separated by SDS–PAGE and autoradiography. The efficiency of protein import was quantified by densitometry of the autoradiography images using the ImageQuant TL program. The level of import into control mitochondria at the indicated time point was set to 100%.

### Redox state analysis of COA7

The oxidation state of COA7 was determined by direct and indirect thiol trapping assays with isolated mitochondria. For the direct thiol trapping assay, mitochondria were solubilized in Laemmli sample buffer either with 50 mM DTT, or 10 mM IAA, or 15 mM AMS (Life Technologies). For the indirect thiol trapping assay, mitochondria were treated with IAA to block free cysteine residues for 10 min at 30°C. The solution was centrifuged at 20,000 *g* for 10 min at 4°C, and mitochondria were resuspended in trehalose buffer that contained 50 mM DTT at 65°C for 15 min to reduce the disulfide bonds. Finally, mitochondria were solubilized in Laemmli buffer with AMS for 30 min at 30°C. The samples were denatured and analyzed by SDS–PAGE and Western blot.

### Redox state analysis of MIA40 — cells

HEK293 cells grown in galactose growth medium were harvested by trypsinization, and proteins were precipitated by addition of trichloroacetic acid (TCA, 9% final). As reference, control cells were pretreated for 10 min at 37°C with 10 mM DTT or 10 mM diamide to reduce or oxidize CPC motive of MIA40, respectively (Erdogan *et al*, 2018). Precipitate was isolated by centrifugation, and the pellet was washed with 5% TCA. The resulting pellet was solubilized in urea buffer (6 M urea, 0.5% SDS, 10 mM EDTA, 200 mM Tris–HCl [pH 7.5]) and aliquoted into samples reflecting approx. $1 \times 10^6$ cells. Samples were precipitated with ice-cold acetone (6:1 vol:vol) and stored at −20°C overnight. After thawing, the precipitate was isolated by centrifugation, solubilized in 2× Laemmli sample buffer supplemented with 1 mM PEG-PCMal, and incubated at 37°C for 30 min. Sample was then mixed with 2× Laemmli sample buffer supplemented with 100 mM DTT (1:1, vol:vol) and denatured at 65°C for 20 min. Proteins were separated on SDS–PAGE followed by Western blotting.

### Redox state analysis of MIA40 — mitochondria

Mitochondria isolated from HEK293 cells were resuspended in RIPA supplemented with 1 mM PEG-PCMal. Mitochondria lysate was incubated at 37°C for 30 min and centrifuged. The supernatant was collected, mixed with 5× Laemmli sample buffer (2:1, vol:vol), and denatured at 65°C for 20 min. As a control, mitochondria were solubilized in RIPA buffer without PEG-PCMal and mixed 5× Laemmli sample buffer with 50 mM DTT. Proteins were separated on SDS–PAGE followed by Western blotting.

### Proliferation in galactose assay

24 h prior to the experiment, human fibroblasts were seeded into six-well format (two wells per experimental variant) at concentration of $0.1 \times 10^6$/well in the growth medium. At the day 0, cells from two wells were detached by trypsinization and counted in a cell counter (Countess II, Life Technologies). Medium in remaining wells was exchanged into DMEM supplemented with 10 mM galactose or 10 mM glucose, 10% (v/v) heat-inactivated fetal bovine serum, 2 mM L-glutamine, 50 μg/ml uridine, 100 μM pyruvate, 100 U/ml penicillin, and 10 μg/ml streptomycin sulfate. After 24 h of culture at 37°C with 5% $CO_2$ (day 1), cells were harvested by

trypsinization and counted. Cell growth was expressed as a fold change of cell count on day 1 vs. day 0.

## Complex IV activity

Approximately $10 \times 10^6$ immortalized fibroblast cells were cultured for 24 h and subjected to proteasome inhibition. The cells were then harvested by trypsinization and resuspended in buffer A (20 mM MOPS/KOH [pH 7.4] and 250 mM sucrose) followed by digitonin treatment (0.2 mg/ml) for 5 min at 4°C. The cell lysate was centrifuged at 5,000 $g$ for 3 min. The supernatant that contained the cytosolic fraction was discarded, and the pellet was resuspended in buffer B (20 mM MOPS/KOH [pH 7.4], 250 mM sucrose, and 1 mM EDTA). The solution was centrifuged at 10,000 $g$ for 3 min, and the pellet that contained permeabilized cells was resuspended in 10 mM potassium phosphate buffer (pH 7.4) and frozen-thawed three times in liquid nitrogen immediately before starting the spectrophotometric assay of complex IV activity (Tiranti *et al*, 1995). 200 μl of 100 μM reduced cytochrome *c* in 50 mM potassium phosphate buffer (pH 7.0) was pipetted into 96-well format plate. The oxidation of cytochrome *c* was initiated by addition of 10 μl of digitonized cell suspension (10 μl potassium phosphate buffer for blank sample), and the decrease in absorbance at 550 nm was recorded for 3 min, which corresponded to linear changes in absorbance. Specific activity of complex IV was calculated by subtracting the absorbance change in the blank sample. The protein concentration of digitonized cell suspension was measured by Bradford assay, and complex IV activity was expressed in nanomoles of cytochrome *c* per minute per milligram of protein.

## Blue-Native PAGE — mitochondria

Isolated mitochondria were solubilized in digitonin buffer (1% [wt/vol] digitonin, 20 mM Tris–HCl, pH 7.4, 50 mM NaCl, 10% [wt/vol] glycerol, 0.1 mM EDTA, 1 mM PMSF) at the concentration of 1 μg of mitochondria/ 1 μl buffer. The mitochondrial suspension was resuspended by gentle pipetting and incubated at 4°C at 15 min. After lysis, the cell suspension was centrifuged at 20,000 $g$ for 15 min at 4°C. The supernatant was transferred to a prechilled Eppendorf tube, and 10× loading dye was mixed by centrifuging at 20,000 $g$ for 15 min at 4°C. The sample was directly applied to 4–13% gradient gel and resolved at 4°C. Protein complexes were transferred to PVDF membranes and immunodecorated with specific antibodies. The High Molecular Weight Calibration Kit for native electrophoresis (Amersham) was used as a molecular weight standard.

## Blue-Native PAGE — digitonized cell lysate

Approximately $5 \times 10^6$ immortalized fibroblasts were harvested by trypsinization and washed two times with ice-cold PBS. The cell pellet was resuspended in 400 μl of cold 8 mg/ml digitonin suspension in PBS and incubated at 4°C for 15 min. Then, the lysate was diluted with ice-cold PBS and centrifuged for 5 min at 10,000 $g$ at 4°C. The pellet was resuspended in 200 μl of solubilization buffer (1.5 M aminocaproic acid, 50 mM Bis-Tris–HCl pH 7.0) containing 20 μl of 10% DDM for 5 min at 4°C. After solubilization, samples were centrifuged at 18,000 $g$ for 30 min at 4°C. The supernatant

was mixed with 20 μl of sample buffer (750 mM aminocaproic acid, 50 mM Bis-Tris–HCl pH 7.0, 0.5 mM EDTA, and 5% Serva Blue G). Samples were directly applied to 4–13% gradient gel and resolved at 4°C. Protein complexes were transferred to PVDF membranes and immunodecorated with specific antibodies. The High Molecular Weight Calibration Kit for native electrophoresis (Amersham) was used as a molecular weight standard.

## Immunofluorescence

HeLa cells or HeLa cells that stably expressed COA7-HA were transfected with different subcompartment markers, namely matrix targeted photoactivatable GFP (mtPAGFP) [matrix], COX8A-DsRed [inner membrane], and TOMM20-DsRed [outer membrane], and fixed with 3.7% formaldehyde after 24 h. The cells were permeabilized (1% Triton X-100 and 0.1% sodium deoxycholate in phosphate-buffered saline [PBS]) and immunostained in blocking buffer (5% goat serum and 0.1% IgG-free BSA in PBS) with anti-TOMM20 (Santa Cruz Biotechnology), anti-SMAC/DIABLO (Abcam), anti-SDHA (Abcam), anti-Aconitase 2 (Abcam), anti-MDH2 (Atlas Antibodies), anti-COA7 (Atlas Antibodies), or anti-HA (Cell Signaling Technology) antibody and the respective secondary antibody (Alexa Fluor 488 or 568, H+L- or IgG-specific, Invitrogen) and mounted with Prolong Diamond (Invitrogen).

## Super-resolution imaging and analysis

HeLa cells were transfected with different subcompartment markers (mtPAGFP [matrix], COX8A-DsRed [inner membrane], and TOMM20-DsRed [outer membrane]) and fixed with 3.7% formaldehyde after 24 h. The cells were permeabilized with 1% Triton X-100 and 0.1% sodium deoxycholate in PBS and immunostained in blocking buffer (5% goat serum and 0.1% IgG-free BSA in PBS) with anti-TOMM20 (Santa Cruz Biotechnology), anti-Smac/Diablo (Abcam), anti-SDHA (Abcam), anti-Aconitase 2 (Abcam), anti-MDH2 (Atlas Antibodies), and the respective secondary antibody (Alexa Fluor 488 or 568, H+L- or IgG-specific, Invitrogen) and mounted with Prolong Diamond. Acquisition was performed using an N-SIM microscope system (Nikon) equipped with a super-resolution Apo TIRF 100× 1.49 NA objective and a DU897 Ixon camera (Andor Technologies). Three-dimensional SIM image stacks were acquired with a Z-distance of 0.15 μm. All of the raw images were computationally reconstructed using the reconstruction slice system from NIS-Elements software (Nikon) while keeping the same parameters. The co-localization analysis was performed using Imaris 9.0 XT software (Bitplane Scientific Software, St. Paul, MN, USA. For each image, the threshold was applied in the same way as the controls, and Pearson's coefficient in the co-localized volume was calculated for each image. The statistical analysis was performed using GraphPad Prism software and one-way analysis of variance (ANOVA) followed by Tukey's multiple comparison post hoc test.

## Mass spectrometry

The affinity-purified eluate fraction was resuspended in 100 μl of 100 mM ammonium bicarbonate buffer, reduced in 100 mM DTT for 30 min at 57°C, alkylated in 55 mM iodoacetamide for 40 min at RT in the dark, and digested overnight with 10 ng/ml trypsin (CAT

NO V5280, Promega) at 37°C. Finally, to stop digestion trifluoroacetic acid was added at a final concentration of 0.1%. Mixture was centrifuged at 4°C, 14,000 *g* for 20 min, to remove solid remaining. MS analysis was performed by LC-MS in the Laboratory of Mass Spectrometry (IBB PAS, Warsaw) using a nanoAcquity UPLC system (Waters) coupled to a LTQ-Orbitrap Velos or Q Exactive mass spectrometer (Thermo Fisher Scientific). The peptide mixture were separated on precolumn (C18, Waters) coupled with nano-HPLC RP18 column (Milford's Waters, 75 μM) using 180 min gradient (0–35% B) solution A: 0.1% TFA in water, B: CAN, 0.1% TFA. The mass spectrometer was operated in the data-dependent MS2 mode, and data were acquired in the m/z range of 300–2,000. Peptides were separated by a 180 min linear gradient of 95% solution A (0.1% formic acid in water) to 35% solution B (acetonitrile and 0.1% formic acid). The measurement of each sample was preceded by three washing runs to avoid cross-contamination. Data were analyzed with the MaxQuant (version 1.5.7.4) platform using mode match between runs (Cox & Mann, 2008). The reference human proteome database from UniProt was used (downloaded ta 2015.11.20). Variable modification was set for methionine oxidation, acetyl at protein n-term, and fixed modification with carbamidomethyl on cysteines. Label-free quantification (LFQ) intensity values were calculated using the MaxLFQ algorithm (Cox *et al*, 2014). Identified proteins were analyzed as follows. Protein abundance was defined as the signal LFQ intensity calculated by MaxQuant software for a protein (sum of intensities of identified peptides of given protein) divided by its molecular weight. Specificity (enrichment) was defined as the ratio of the protein LFQ intensity measured in the bait purification (WT) to background (FP) level. LFQ for proteins not detected was arbitrarily set to 1 for calculation reasons.

### Molecular modeling

The structure of COA7 (UniProt id: Q96BR5) was modeled using the Yasara Structure 17.1.28 package based on three Protein Data Bank (PDB) structures (1OUV, 4BWR, and 1KLX) as templates (Luthy *et al*, 2002, 2004; Urosev *et al*, 2013). These PDB structures were selected automatically based on the combination of the blast E-value, sequence coverage, and structure quality. For each template, up to five alternate alignments with the target sequence were used, and up to 50 different conformations were tested for each modeled loop. The resulting models were evaluated according to structural quality (dihedral distribution, backbone, and side-chain packing). The model with the highest score of these that covered the largest part of the target sequence was used as the template for a hybrid model that was further iteratively improved with the best fragments (e.g., loops) that were identified among the highly scored single-template models.

### Molecular dynamics

Molecular dynamics analysis was performed using the Yasara structure with a standard Yasara2 field (Krieger *et al*, 2009). The initial structures were solvated in cubic boxes of water molecules, the dimensions of which allowed at least 20 Å separation/distance between the protein and box border. The general fold was initially preserved by weak distance constraints that were applied for all

H-bonds that were identified in the helical regions. The molecular dynamics simulations in the isothermal–isobaric ensemble (NPT) were performed for an initial 5 ns with fixed backbone atoms (N, Cα, and C) to enable optimization of the side chains. During the next 25 ns, the system was released to evolve, and molecular dynamics snapshots were taken every 10 ps.

The conformation and flexibility of all non-helical regions were analyzed by the time evolution of phi and psi backbone angles, and are presented as Ramachandran plots. All of these analyses were performed with the R 3.3.0 package (www.r-project.org) using custom-made scripts.

### Statistics

All numerical data are expressed as a mean ± SEM (standard error) except of the quantification of immunofluorescence co-localization, which is expressed as a mean ± SD (standard deviation). The exact *P*-values and the number of biological repeats (*n*) for each experiment are indicated in the figure legends. If not stated otherwise, the numerical data were a subject to the Shapiro–Wilk test for normal distribution, and significance was calculated using the two-tailed Student's *t*-test. No randomization protocol was used. No blinding protocol was used.

**Expanded View** for this article is available online.

### Acknowledgements

We thank Anabel Martinez Lyons, MRC Mitochondrial Biology Unit, University of Cambridge, for providing the immortalized patient fibroblasts, Ben Hur Mussulini, Laboratory of Mitochondrial Biogenesis, University of Warsaw, for help in assaying respiratory chain activity and Aleksandra Gosk for help in CRISPR ID analysis. This work was financed by the National Science Centre, Poland (NCN; grant no. NCN 2012/05/B/NZ3/00781 and NCN 2015/19/B/NZ3/03272); Ministerial funds for science within the Ideas Plus program (000263 in 2014–2017); "Regenerative Mechanisms for Health" project MAB/2017/2, carried out within the International Research Agendas programme of the Foundation for Polish Science co-financed by the European Union under the European Regional Development Fund; Core Grant from the MRC (Grant MC_UU_00015/5) (to MZ); ERC Advanced under Grant FP7-322424 and NRJ-Institut de France Grant (to MZ); ERC (ERC-AdG MITRAC No. 339580) (to PR); SFB1190 (P13) (to PR); Foundation of Polish Science TEAM TECH CORE FACILITY/2016-2/2 Mass Spectrometry of Biopharmaceuticals—improved methodologies for qualitative, quantitative, and structural characterization of drugs, proteinaceous drug targets, and diagnostic molecules. The biobank "Cell Line and DNA Bank of Genetic Movement Disorders and Mitochondrial Diseases", a member of the Telethon Network of Genetic Biobanks (project no. GTB12001), funded by Telethon Italy, and the EuroBioBank Network provided the control and patient-derived fibroblast specimens.

### Author contributions

KM, MW, and AC designed the study. KM performed most of the experiments and evaluated the data together with MW and AC. CB, ZB, and MW performed experiments. EF-V, MW, PS, MD, and SD provided reagents and methods or created cell lines with modified gene expression. JP performed the molecular modeling and simulation studies. DC and MD (IBB) performed the mass spectrometry analysis. MW, PR, MZ, and AC supervised the study and interpreted the results. KM, MW, CB, JP, and AC prepared the figures and wrote the manuscript. All of the authors commented on the manuscript.

## The paper explained

### Problem

Mitochondrial diseases are a clinically heterogeneous group of genetic disorders that are characterized by dysfunctional mitochondria. To date, the treatment for such diseases consists mostly of metabolite supplementation, which is only a symptomatic treatment and does not treat the underlying pathological condition.

### Results

The present study characterized a protein, COA7, previously shown to be involved in the assembly and function of respiratory chain complexes, as an intermembrane space protein and a new substrate of MIA40. We also characterized pathogenic disease-causing COA7 variants and found that they were imported to mitochondria more slowly than wild-type COA7. Consequently, both these mutant proteins were mislocalized to the cytosol and degraded by the proteasome. Proteasome inhibition resulted in an increased mitochondrial localization of mutant proteins. Inhibition of the proteasome led to an increase in the mitochondrial localization of mutant COA7 variants and rescued mitochondrial levels of other affected mitochondrial proteins. We also found that inhibition of the proteasome rescued a defect of respiratory complex IV in COA7-deficient patient fibroblast.

### Impact

The present study identified an important role for the proteasome in degradation of mutant mitochondrial proteins under pathological conditions. Inhibition of the proteasome may be beneficial for patients who suffer from mitochondrial diseases that are characterized by a lower amount of mitochondrial proteins. We raise the possibility that proteasome inhibitors (e.g., bortezomib and carfilzomib) that are already used clinically for cancer therapy can restore the levels of mitochondrial proteins that despite being mutated conserve some functionality when allowed to reach the right compartment. With currently limited therapeutic options for treating mitochondrial diseases, this strategy may be promising for the mitochondrial diseases, in which mitochondrial protein depletion is observed and contributes to lowering respiratory complex activity.

## Conflict of interest

The authors declare that they have no conflict of interest.

## For more information

(i)  Mitochondrial complex IV deficiency: https://www.omim.org/entry/220110

(ii)  *COA7* gene: https://www.omim.org/entry/615623.

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
