## [Review Process File · EMBO Molecular Medicine]

Inhibition of proteasome rescues a pathogenic variant of respiratory chain assembly factor COA7

Karthik Mohanraj, Michal Wasilewski, Cristiane Benincá, Dominik Cysewski, Jarosław Poznanski, Paulina Sakowska, Zaneta Bugajska, Markus Deckers, Sven Dennerlein, Erika Fernandez-Vizarra, Peter Rehling, Michal Dadlez, Massimo Zeviani, and Agnieszka Chacinska

Review timeline:	Submission date:	20 July 2018
	Editorial Decision:	23 August 2018
	Revision received:	8 January 2019
	Editorial Decision:	6 February 2019
	Revision received:	15 February 2019
	Accepted:	18 February 2019

Editor: Céline Carret

Transaction Report:

1st Editorial Decision

23 August 2018

Thank you for the submission of your manuscript to EMBO Molecular Medicine. We have now heard back from the three referees whom we asked to evaluate your manuscript.

You will see from the comments below that the referees find the paper interesting and carefully performed and all provided enthusiastic reports. Ref.1 and 2 mainly have included suggestions and questions to answer to improve conclusiveness and strengthen the paper, which we would encourage you to address. Please also pay attention to formatting/display issues, i.e. providing an explanatory cartoon for ex. sounds like a good idea.

We would therefore welcome the submission of a revised version within three months for further consideration. Please note that EMBO Molecular Medicine strongly supports a single round of revision and that, as acceptance or rejection of the manuscript may depend on another round of review, your responses should be as complete as possible.

I look forward to receiving your revised manuscript.

***** Reviewer's comments *****

Referee #1 (Comments on Novelty/Model System for Author):

The technical quality is very high. However, description of experiments and statistical analysis needs to be strongly improved

Novelty of the latter manuscript part (Fig. 8-9) is high.

Referee #1 (Remarks for Author):

The MIA machinery for many proteins drives protein import into the mitochondrial intermembrane space. Mohanraj and colleagues describe here the characterisation of a novel MIA40/CHCHD4 substrate, called RESA1/COA7. In figures 1-6 the interaction with MIA40 and the oxidation of RESA1 is extensively characterized demonstrating that RESA1 indeed behaves in many aspects like classical MIA substrates. In the last figures, an interesting RESA1 mutant that is hampered in mitochondrial import is introduced. Cytosolic accumulation of the RESA1 mutant prompts its proteasomal degradation. Inhibition of this degradation appeared to partially rescue both, mitochondrial localization of RESA1 and complex IV activity (for which RESA1 is an assembly factor).

This is a well-written and data-rich manuscript. Especially the characterisation of RESA1 as MIA40 substrate is comprehensive. However, this part is also less exciting than the findings presented later in the manuscript. This is mainly because Petrunaro and colleagues already found RESA1 (as C1orf163) in their interactome of MIA40 as an interaction partner that coprecipitates under denaturing conditions indicating disulfide-linkage between MIA40 and RESA1 (Cell Metab, 2015), and Kozjak-Pavlovic and colleagues described its function and the consequences of its absence already in 2014 (JMB, 2014). Still, the mechanistic unravelling of the precise import and oxidation mechanism is of high value.

The last figures present the most exciting and novel insights into the biology of MIA-dependent import. I feel however that this part is underdeveloped, and that it would require some additional work to render this manuscript suitable for publication.

Major points

Figure 7-9 describe a positive effect of proteasomal inhibition on mitochondrial RESA1 mutant levels and complex IV activity. This part should be extended to better understand the underlying mechanisms and the physiological consequences of the proteasomal inhibition. Moreover, the experimental procedures remain at many places nebulous especially with respect to the cell lines (transient expression, stable cell lines - single or multiple copies) used. The following points are a list of suggestions/questions that I think are important to address to improve this part of the manuscript

1/ Figure 7: Please provide a scheme of the RESA1 disease mutant. The patient carries both mutations. Is there crosstalk between the two RESA1 mutants? Are experiments depicted in figure 7 performed in the presence of endogenous RESA1? Are RESA1 mutants dominant-negative, i.e. are complex IV levels and activity changed upon mutant expression? Are the depicted experiments performed with transient expression or with stable cell lines? If it were transient expression, how can you be sure that you always have similar expression levels (just as an example: compare e.g. Figures 7A and B, expression levels of the two mutants with respect to each other differ)? This point is especially important as later experiments in the presence of MG132 indicate changes in protein levels.

2/ Figure 8: Please provide quantifications. It remains otherwise unclear whether only the levels of RESA1 in both cytosol and IMS increase upon MG132 treatment or whether the ratio is tilted towards the IMS. If at all, the ratio seems to shift towards the cytosol (8F).

3/ Figure 8D: Why does MG132 only have an effect in the absence of CHX? If the recognition for degradation were an early event, than the authors should perform radioactive pulse-chase experiments

4/ Figure 8D: MG132 has no effect on endogenous RESA1, yet it exerts an effect on overexpressed wild type RESA1 (almost as strong as with the RESA1 mutants). How high are overexpression levels? The authors should titrate RESA1-WT levels to endogenous amounts and then confirm that they do not observe effects of MG132 treatment.

5/ Figure 8: Does RESA1 become ubiquitinated upon MG132 treatment?

6/ Figure 9A: Why does the HSP70 signal disappear without proteasomal inhibitor treatment?

7/ Figure 9C,D: what is the statistical reasoning for using 'standard error' and not 'standard deviation'? On what was the normalization, i.e. how do you know that mitochondrial isolation worked equally well? Unfortunately, the actual complex IV activity assay was described only poorly. The description should be improved.

8/ Figure 9. Respiratory chain activity upon MG132 treatment: This part should be extended by BN-PAGE analysis (is there more assembled complex IV present upon MG132 treatment), oxygen consumption assays, viability assays on galactose etc.. Moreover, comparison with control fibroblasts is missing.

9/ There appear to be certain 'redundancies' in the first figures. Many panels show in orthogonal approaches the interaction between MIA40 and RESA1. These figures could be condensed to allow expansion of the latter figures describing the interesting 'proteasomal effect'.

Minor points:

1/ The labelling of figures/extent of experimental description in figure legends (what has been done?, MW marker, which kind of IP, how was MIA40 expressed in the different figures: transient, stable cell lines, stable inducible cell lines, etc. a heavy overexpression might result in mislocalization and influence interpretation of results etc) is underdeveloped. This has to be improved to enable understanding of the performed experiments. Likewise, n numbers and quantifications of experiments are missing.

2/ The nomenclature in the field is somewhat of a mess. I would ask the authors to mention that MIA40 is also referred to as CHCHD4, and that ALR is the human homolog of the yeast Erv1.

3/ Figure 1B: Please provide additional immunoblots against more classical substrates of MIA40

4/ Figure 1A: Provide the proteomics data in full, e.g. as an excel file in the SI or as a link to a database.

5/ Figure 2D shows a very uneven expression of MIA40 variants. Is this due to transient expression?

6/ Figure 3A: The scheme is somewhat hard to grasp. Could the authors provide a better version?

7/ Figure 4D: Isolated mitochondria are not a good model for the determination of cysteine redox states. How can the authors ensure preservation of the endogenous redox state? Moreover, to present the data, the exposure should be varied and quantifications of the ratio 'ox/red' should be provided.

8/ Figure 6A: Provide sequencing data for the MIA40 CRISPR clones.

9/ Figure 6B: Why is MIA40 almost completely gone if only one allele is affected?

Referee #2 (Comments on Novelty/Model System for Author):

The authors use in vitro import assays in combination with assays in cultured cells. This allows them to tackle mechanistic and physiological questions.

Referee #2 (Remarks for Author):

This manuscript characterizes COA7 (RESA1) as an IMS protein and a non-canonical substrate of MIA40. Some of the authors had previously reported a matrix localization of the protein but here its IMS residence and the involvement of MIA40 for its import are undoubtedly demonstrated. The protein is biomedically relevant because mutations in the human gene have been associated with mitochondrial leukoencephalopathy associated with mitochondrial respiratory chain complex IV deficiency. The authors identified the COA7 mutations as responsible for import failure, which leads to retention of the newly synthesized protein in the cytoplasm. However, the protein does not accumulate in this compartment because is actively degraded by the proteasome. The authors elegantly show that overexpression of mutant COA7 or inhibition of the proteasome with either MG132 or other clinically approved proteasome inhibitors restores COA7 levels in mitochondria and also its activity in complex IV assembly. Therefore, the authors suggest that proteasome inhibition may be a new venue to combat mitochondrial diseases associated with poor mitochondrial protein import or excessive degradation by the proteasome.

The manuscript is technically and conceptually sound, and appropriate for publication in EMBO Molecular Medicine.

I have only two requests to improve the manuscript:

1- The experiments presented do not completely exclude the possibility that mutant RESA1 is not degraded by mitochondrial proteases. The authors should silence some of these proteases (e.g. AAA proteases) and test whether the protein is still degraded with a similar efficiency.

2- The authors suggest that clinically approved proteasome inhibitors, such as bortezomib, may be applied as therapeutic agents to combat at least a subset of mitochondrial disorders. The authors should discuss the potential side effects on mitochondria and other organelles.

Minor points:

1- Page 10: "TIMM8A (CX9C) and COX19 (CX3C)" should be "TIMM8A (CX3C) and COX19 (CX9C)"

2- I strongly suggest the authors to use COA7 to refer to the protein. The use of alternative names only serves to confuse the literature.

Referee #3 (Remarks for Author):

This manuscript reports a careful analysis of the biogenesis of the mitochondrial respiratory chain assembly factor 1 (RESA1). RESA1 has been linked to mitochondrial leukoencephalopathy and complex IV deficiency. The authors show that RESA1, which contains 13 cysteine residues, is an unusual substrate of the mitochondrial intermembrane space assembly (MIA) system. In a remarkably complete characterization, they elucidated the molecular mechanisms of import of RESA1 into the mitochondrial intermembrane space, the interaction with Mia40 and disulfide bond formation. Importantly, the mitochondrial import of pathogenic mutant versions of RESA1 is slower than that of wild-type RESA1 and proteins accumulating in the cytosol are degraded by the proteasome. Using patient-derived fibroblasts, the authors discovered that inhibition of the proteasome rescued the localization of the mutant RESA1 to mitochondria and the activity of complex IV.

This paper by leading experts of the field is of technically very high quality and written very well. It provides exciting novel findings on the role of the proteasome in the pathogenesis of mitochondrial diseases and opens the way for new therapeutic approaches by using clinically approved proteasome inhibitors.

I have only a few minor comments on this exciting paper.

1. The authors provide a complete characterization of the biogenesis of wild-type and mutant RESA1 with important medical implications. It would be helpful for the general readership to present a cartoon of the import pathway of RESA1 and the role of the proteasome, e.g. in the last figure.

2. Suggestions for corrections (indicated in CAPITAL letters):

- Page 3, middle: ... a precursor protein enters mitochondria via THE translocase of the outer membrane ...

- Page 8, line 3 from the bottom: These observations suggest that among the 13 cysteine residues, LIKELY 10 are involved in disulfide bonds, ...

- Page 17, middle: Treatment of patient fibroblasts with PROTEASOME inhibitors led to approximately ...

1st Revision - authors' response

8 January 2019

Referee #1 (Remarks for Author):

The MIA machinery for many proteins drives protein import into the mitochondrial intermembrane space. Mohanraj and colleagues describe here the characterization of a novel MIA40/CHCHD4 substrate, called RESA1/COA7. In figures 1-6 the interaction with MIA40 and the oxidation of RESA1 is extensively characterized demonstrating that RESA1 indeed behaves in many aspects like classical MIA substrates. In the last figures, an interesting RESA1 mutant that is hampered in mitochondrial import is introduced. Cytosolic accumulation of the RESA1 mutant prompts its proteasomal degradation. Inhibition of this degradation appeared to partially rescue both, mitochondrial localization of RESA1 and complex IV activity (for which RESA1 is an assembly factor).

This is a well-written and data-rich manuscript. Especially the characterisation of RESA1 as MIA40 substrate is comprehensive. However, this part is also less exciting than the findings presented later

in the manuscript. This is mainly because Petrunaro and colleagues already found RESA1 (as C1orf163) in their interactome of MIA40 as an interaction partner that coprecipitates under denaturing conditions indicating disulfide-linkage between MIA40 and RESA1 (Cell Metab, 2015), and Kozjak-Pavlovic and colleagues described its function and the consequences of its absence already in 2014 (JMB, 2014). Still, the mechanistic unravelling of the precise import and oxidation mechanism is of high value.

The last figures present the most exciting and novel insights into the biology of MIA-dependent import. I feel however that this part is underdeveloped, and that it would require some additional work to render this manuscript suitable for publication.

Major points

Figure 7-9 describe a positive effect of proteasomal inhibition on mitochondrial RESA1 mutant levels and complex IV activity. This part should be extended to better understand the underlying mechanisms and the physiological consequences of the proteasomal inhibition. Moreover, the experimental procedures remain at many places nebulous especially with respect to the cell lines (transient expression, stable cell lines - single or multiple copies) used. The following points are a list of suggestions/questions that I think are important to address to improve this part of the manuscript

1/ Figure 7:

Please provide a scheme of the RESA1 disease mutant.

A: We have included a schematic presentation of mutated RESA1/COA7 in Figure 6A of the revised manuscript.

The patient carries both mutations. Is there crosstalk between the two RESA1 mutants?

A: In patient fibroblasts we observed the decreased steady state levels of the exon2^d mutant compared to Y137C and wild-type. Similarly, in HEK293 cells when mutant versions were overexpressed separately, the exon2^d mutant was expressed to a lower level than the Y137C mutant. Therefore we think that the observed disproportion in protein levels is unlikely to result from a crosstalk between both mutant variants of RESA1/COA7. We have introduced this notion in Results section when we first describe levels of RESA1/COA7 in mt4229i cells (Fig EV3). Moreover, in response to the reviewer's question about possible dominant-negative effects of mutant proteins, we verified effects of mutant expression on cytochrome c oxidase in the presence of a wild-type protein and found no influence (please see the response below).

Are experiments depicted in figure 7 performed in the presence of endogenous RESA1?

A: All the experiments presented in Figure 7 (Figure 6 in the revised manuscript) were performed in the presence of native RESA1/COA7. The expression of wild-type and mutant RESA1/COA7 was obtained by transient expression. We have clarified this point both in figure legends and in the corresponding text of Results section.

Are RESA1 mutants dominant-negative, i.e. are complex IV levels and activity changed upon mutant expression?

A: Our approach to address reviewer's comment was to overexpress separately the mutated forms of RESA1/COA7 in HEK293 cells carrying wild type alleles of the protein. We then verified steady state levels of subunits of cytochrome c oxidase and the activity of the complex (new data included in Appendix Fig S5A and B). We did not find any negative effects of mutants' overexpression upon either composition or activity of cytochrome c oxidase. We conclude that mutants do not impinge a dominant-negative effect in the background of wild-type protein. This notion is further supported by the fact that both parents of the patient, who carry single alleles of pathogenic variants of RESA1/COA7, do not present any symptoms of the disease (Martinez Lyons et al, 2016).

Are the depicted experiments performed with transient expression or with stable cell lines? If it were transient expression, how can you be sure that you always have similar expression levels (just as an example: compare e.g. Figures 7A and B, expression levels of the two mutants with respect to each other differ)? This point is especially important as later experiments in the presence of MG132 indicate changes in protein levels.

A: In the experiments presented in the Figure 7 (Figure 6 in the revised manuscript) we used transient transfection in order to obtain overexpression of proteins in HEK293 cells. We have now clarified this point both in figure legends and in the Results section.

We agree with the reviewer's remark that transient transfection induces more variable levels of expression than those observed e.g. in stable cell lines. In order to prevent misinterpretation we standardized transfection conditions i.e. cell densities, timing of experiments and amounts of DNA and transfectant used. We have included the detailed description of this procedure in the Appendix section. Our method of transfection reproducibly yielded lower levels of mutant protein expression than the wild-type RESA1/COA7. In the majority of experiments we also observed the lower levels of the exon2^d mutant as compared with the Y137C mutant. In few experiments we saw slight variability in the exon2^d mutant expression, yet expression of the Y137C mutant was very reproducible. The reviewer suggests that our interpretation of the apparent rescue of mutant proteins by proteasome inhibition (Figure 7 D and F of the revised manuscript) may be hindered by unequal transfection. In our opinion, there is a minimal chance that this may have influenced our interpretation in this particular experimental setting. However, our conclusions are supported by the experiments on patient fibroblasts wherein we demonstrated that proteasome inhibition rescued also mutant proteins of physiological abundance (Figure 8A and B of the revised manuscript).

2/ Figure 8: Please provide quantifications. It remains otherwise unclear whether only the levels of RESA1 in both cytosol and IMS increase upon MG132 treatment or whether the ratio is tilted towards the IMS. If at all, the ratio seems to shift towards the cytosol (8F).

A: This comment tackles an important aspect of our discourse that obviously was inadequately explained in the manuscript. Inhibition of the proteasome increases an overall abundance of RESA1/COA7 mutants in the cell. In fact a substantial portion of the mutant protein, especially during transient expression, is present in the cytosol. Our data demonstrates however that in parallel the mitochondrial content of mutant proteins is also increased. Below we present quantification of localization of exon 2 deletion mutant with and without treatment with MG132. It demonstrates that indeed the protein abundance in the cytosol increases more than in mitochondria. However, it is the mitochondrial content of mutant protein that is beneficial to mitochondria function irrespective to changes in the ratio between cytosolic and mitochondrial fractions. In the submitted version of the manuscript we used a phrase "... more interestingly a larger portion of the mutant proteins was localized in mitochondria (Fig 8F, lanes 3, 7, 10, 14).", which could suggest that we meant "a fraction of mutant protein". Therefore in the revised manuscript we modified this sentence to avoid this misleading trait: "Indeed, in the presence of MG132 mediated proteasome inhibition, the levels of RESA1 mutant proteins increased in the cytosol, which was paralleled by an increased mitochondrial content of the mutant proteins (Fig 7F, lanes 3, 7, 10, 14)".

[Unpublished data removed upon the authors' request.]

3/ Figure 8D: Why does MG132 only have an effect in the absence of CHX? If the recognition for degradation were an early event, than the authors should perform radioactive pulse-chase experiments

A: In order to address this comment of the reviewer, we performed a pulse-chase experiment, in which we inhibited proteasome either during the radiolabeling of newly synthesized proteins (A) or directly after the labelling (B) or 1h after the labelling (C) (please see a scheme below). The chase was finalized by the affinity purification of overexpressed RESA1/COA7-Y137C_{HIS}. When MG132 was applied 1h after the end of the labelling (C) we could not observe any increase in RESA1/COA7 in the eluate (line 10). This is consistent with the interpretation that RESA1/COA7 is protected from proteasomal degradation shortly after the synthesis when it is efficiently imported into mitochondria. Unfortunately, at earlier times of MG132 treatment (A and B) we observed a significant decrease of protein labelling (load fraction), which reflected a temporal decrease of translation caused by MG132. This side effect of proteasome inhibition has been previously described (Jiang & Wek, 2005; Wu et al, 2009). Therefore we could not conclude about the early effects of proteasomal inhibition on the stability of RESA1/COA7-Y137C_{HIS}. Taking this result into consideration we have milder our conclusions in the Results section and stated that: "In contrast under active translation mutant COA7 were degraded by proteasome, while wild-type protein was only marginally affected suggesting that proteins with slower rate of import to mitochondria were sensitive to proteasome-mediated degradation (Fig 7D, lanes 4 and 5)." We also modified the Discussion section accordingly.

4/ Figure 8D: MG132 has no effect on endogenous RESA1, yet it exerts an effect on overexpressed wild type RESA1 (almost as strong as with the RESA1 mutants). How high are overexpression

levels? The authors should titrate RESA1-WT levels to endogenous amounts and then confirm that they do not observe effects of MG132 treatment

A: *In fact, we originally assumed that the influence of proteasome inhibition on the transiently expressed wild-type RESA1/COA7 was a result of protein overproduction. In order to address this issue we titrated down transfection by decreasing the load of DNA/transfectant complexes on the cells (Appendix Fig S4C). We then checked the effect of MG132 upon the overexpressed protein and found that MG132 did not stabilize overexpressed RESA1/COA7 when the load of plasmid DNA was decreased to 1 µg per 60cm² (Appendix Fig S4C). This is consistent with our original assumption that proteasomal degradation of the wild-type RESA1/COA7 results from its overproduction and inefficiency to be imported into mitochondria. However this result does not influence the core conclusions of the manuscript as the effect of proteasome inhibition on mutant RESA1/COA7 was confirmed in patient fibroblasts.*

5/ Figure 8: Does RESA1 become ubiquitinated upon MG132 treatment?

A: *In order to verify whether RESA1/COA7 can undergo ubiquitination we overexpressed ubiquitin tagged with His tag together with COA7_{FLAG} and then performed affinity purification of ubiquitin in the presence of MG132 (Fig EV3A). We purified various species of RESA1/COA7 corresponding to the protein modified with ubiquitin chains of different lengths. At the same time we could not co-purify the native RESA1/COA7, which is consistent with our former interpretation that RESA1/COA7 is subject to proteasome degradation only when the efficiency of import to mitochondria is impaired by mutation in the protein itself or by an increased protein supply in the cytosol due to the overexpression.*

6/ Figure 9A: Why does the HSP70 signal disappear without proteasomal inhibitor treatment?

A: *In the figure 9A (figure 8A of the revised manuscript) we present changes in protein levels following treatments with various inhibitors of proteasome. As it is mentioned in the manuscript HSP70 is known to accumulate in response to proteotoxic stress evoked by inhibition of proteasome, which explains a drastic difference of HSP70 levels between DMSO and inhibitor treated samples (Kim et al, 1999; Awasthi & Wagner, 2005). We included HSP70 Western blot in the previously submitted version of the manuscript as an additional proof that concentration of inhibitors used were effective.*

7/ Figure 9C,D: what is the statistical reasoning for using 'standard error' and not 'standard deviation'? On what was the normalization, i.e. how do you know that mitochondrial isolation worked equally well? Unfortunately, the actual complex IV activity assay was described only poorly. The description should be improved.

A: *We chose standard error to describe our results following general guidelines summarized in (Sullivan et al, 2016). Standard error of the mean is used to express variability of the estimated mean while standard deviation expresses variability in a measure among experimental unit. Therefore standard error should be used to compare groups and standard deviation to describe the distribution of observations measured in the study.*

We apologize for the flows in the description of technical aspects of complex IV activity assay. We introduced more elaborate description of the method in the Materials and Methods section. We used digitonin to permeabilize fibroblasts with as described in (Tiranti et al, 1995). This procedure is much simpler than isolation of mitochondria, which involves homogenization and several steps of centrifugation. Based on our experience this method yields reproducible cellular preparations. The raw activity of complex IV was recalculated in relation to protein level in the sample. Data presented in the Figure 9A, B and C are normalized to control samples obtained from DMSO treated fibroblasts. The normalization to the DMSO treated control was necessary as over time we observed some variability in absolute activity of complex IV. This was most probably due to various batches of digitonin used over time. Commercially available digitonin is a natural plant extract, which contains variable amount of impurities and therefore the activity/quality of different batches may vary.

8/ Figure 9. Respiratory chain activity upon MG132 treatment: This part should be extended by BN-PAGE analysis (is there more assembled complex IV present upon MG132 treatment), oxygen consumption assays, viability assays on galactose etc.. Moreover, comparison with control fibroblasts is missing.

A: We have included comparison of complex IV activity in patient and control fibroblasts, which demonstrates a significant decrease of activity in mt4229i cells (Fig 9A). Following the suggestion of the reviewer, we also analyzed assembly of respiratory chain complexes by BN-PAGE and Western blotting. We found that the main difference between control and patient fibroblasts was expressed in the decrease of supercomplexes as demonstrated with antibodies against complex I (NDUFS1), complex III (UQCRI) and complex IV (COX4 and COX6A) (Fig 9D). In order to further explore the involvement of COA7 in the observed phenomenon we overexpressed wildtype and COA7-Y137C in patient fibroblasts and verified assembly of respiratory chain complexes via BN-PAGE and Western blotting (Fig 9E). In these experiments we used DDM as a solubilizing agent, which in our hands allowed to differentiate between two types of supercomplexes – CI+CIII₂ and CI+CIII₂+CIV. Overexpression of wildtype and mutant COA7 resulted in an increased levels of both types of supercomplexes. This is an interesting observation suggesting that COA7 may in fact influence respiratory complexes beyond complex IV only. This result parallels original report on COA7, in which authors found that RNAi against COA7 affected multiple respiratory chain complexes (Kozjak-Pavlovic et al, 2014). We then verified how treatment with bortezomib affects levels of supercomplexes in patient fibroblasts. We observed an increased presence of supercomplexes in bortezomib treated samples (Fig 9F). This supports our conclusions that mutant variants of COA7, at least COA7-Y137C, when allowed to accumulate in mitochondria can restore biogenesis of respiratory chain .

In addition we performed the proliferation assay on patient and control fibroblasts grown in galactose medium (Appendix Fig S5C and D). Substitution of glucose with galactose forces mammalian cells to depend on oxidative phosphorylation as a main source of ATP. Both cell lines grew slower in galactose medium as compared to glucose medium. We then verified how bortezomib influenced cell proliferation in galactose medium. Concentrations of bortezomib used to increase the assembly of respiratory supercomplexes and activity of complex IV caused a significant decrease in growth rate (data not shown). This is in agreement with a well known role of ubiquitin-proteasome system in the cell cycle (Bassermann et al, 2014). In order to avoid toxic effects of bortezomib we decreased the concentration to 2,5 nM and 1 nM, which was close to the lowest concentration of the drug that was still resulting in increased ubiquitination of proteins. In these conditions 2,5 nM bortezomib treatment for 24 h slightly increased cell growth in patient fibroblasts, yet the effect was statistically insignificant.

9/ There appear to be certain 'redundancies' in the first figures. Many panels show in orthogonal approaches the interaction between MIA40 and RESA1. These figures could be condensed to allow expansion of the latter figures describing the interesting 'proteasomal effect'.

A: In order to render the manuscript more concise we combined Figures 1 and 2. Additionally panels that contained data obtained by orthogonal approaches were moved to Expanded View and Appendix sections.

Minor points:

1/ The labeling of figures/extent of experimental description in figure legends (what has been done?, MW marker, which kind of IP, how was MIA40 expressed in the different figures: transient, stable cell lines, stable inducible cell lines, etc. à heavy overexpression might result in mislocalization and influence interpretation of results etc) is underdeveloped. This has to be improved to enable understanding of the performed experiments. Likewise, n numbers and quantifications of experiments are missing.

A: We have revised figure legends to identify all the weaknesses of description of experimental design. We introduced information requested by the reviewer.

2/ The nomenclature in the field is somewhat of a mess. I would ask the authors to mention that MIA40 is also referred to as CHCHD4, and that ALR is the human homolog of the yeast Erv1.

A: Indeed there is a certain level of inconsistency in the nomenclature referring to the MIA pathway. We use the term MIA40 instead of CHCHD4 because it relates to function of the protein rather than to its structural features and also it corresponds to the name of yeast ortholog where the protein and the pathway was first described. We agree with the reviewer that we should include the latter term and this has been introduced to the revised manuscript when MIA40 is mentioned for the first time. Analogically we introduced a term GFER, which is a systematic name for ALR, and information about homology of ALR to yeast Erv1.

3/ Figure 1B: Please provide additional immunoblots against more classical substrates of MIA40.

A: We were able to demonstrate interactions with TIMM13 and TIMM10B (classical CX₃C substrates of MIA pathway) and this result was incorporated into the revised manuscript as Figure EV1A. Interaction between MIA40 and its substrates is very transient and thus strong antibodies are required to demonstrate it via affinity purification and Western blotting. Unfortunately most of antibodies against MIA40 substrates in our possession are quite weak and thus we could not demonstrate interaction of MIA40 with other precursor proteins.

4/ Figure 1A: Provide the proteomics data in full, e.g. as an excel file in the SI or as a link to a database.

A: We have included proteomics data in excel file format in the Source Data 1.

5/ Figure 2D shows a very uneven expression of MIA40 variants. Is this due to transient expression?

A: Data presented in figure 2D (figure 1D in the revised manuscript) refer to Flp-In T-REx 293 cells. These cells carry one copy of a cassette in the genome that allows for introducing a gene of interest, selection of stable clones and on-demand induction of expression with tetracycline. We have generated several clones, which express wild-type or mutant MIA40 tagged with a FLAG tag. Reproducibly C55S and SPS mutants show lower expression than wild-type and C55S mutant. We do not fully understand the source of this phenomenon. We have added this information in the Materials and Methods section.

6/ Figure 3A: The scheme is somewhat hard to grasp. Could the authors provide a better version?

A: We have simplified the scheme to make it more approachable (Figure 2A of the revised manuscript).

7/ Figure 4D: Isolated mitochondria are not a good model for the determination of cysteine redox states. How can the authors ensure preservation of the endogenous redox state? Moreover, to present the data, the exposure should be varied and quantifications of the ratio 'ox/red' should be provided.

A: Indeed, a procedure of yeast mitochondria isolation increases average oxidation of cysteine residues (Topf et al, 2018). In order to address this issue more accurately we performed redox analysis in total cell extracts. In these experiments the redox state of cysteines was conserved by trichloroacetic acid very early during the procedure. We have found that native redox state of MIA40 was in fact more reduced than in isolated mitochondria. However the new approach did not change our conclusion about the influence of RESA1/COA7 overexpression on MIA40 redox state as it was equal in control cells and cells overexpressing RESA1/COA7. In the revised manuscript we incorporated the whole cell approach as more reliable (Figure 3D of the revised manuscript) and we moved the data obtained from the isolated mitochondria to Appendix Fig S2D.

8/ Figure 6A: Provide sequencing data for the MIA40 CRISPR clones.

A: We included sequencing data in the Source Data 2 and 3.

9/ Figure 6B: Why is MIA40 almost completely gone if only one allele is affected?

A: The reviewer is correct to point out a decrease of wild-type MIA40 in the CRISPR clone (figure 5D of the revised manuscript). The mutant allele of MIA40 lacks the CPC motif, which is responsible for substrate binding and oxidation. In mammalian cells MIA40 is imported to mitochondria via the MIA pathway. We suspect that possessing one allele of protein, which cannot actively support MIA pathway, renders entire pathway less effective. In this sense the mutant allele would exert a dominant negative effect. We have introduced this information in the Results section when we refer to Figure 5D.

References

- Awasthi N, Wagner BJ (2005) Upregulation of heat shock protein expression by proteasome inhibition: an antiapoptotic mechanism in the lens. *Invest Ophthalmol Vis Sci* 46: 2082-2091
- Bassermann F, Eichner R, Pagano M (2014) The ubiquitin proteasome system - implications for cell cycle control and the targeted treatment of cancer. *Biochim Biophys Acta* 1843: 150-162
- Jiang HY, Wek RC (2005) Phosphorylation of the alpha-subunit of the eukaryotic initiation factor-2 (eIF2alpha) reduces protein synthesis and enhances apoptosis in response to proteasome inhibition. *J Biol Chem* 280: 14189-14202

- Kim D, Kim SH, Li GC (1999) Proteasome inhibitors MG132 and lactacystin hyperphosphorylate HSF1 and induce hsp70 and hsp27 expression. *Biochem Biophys Res Commun* 254: 264-268
- Kozjak-Pavlovic V, Prell F, Thiede B, Gotz M, Wosiek D, Ott C, Rudel T (2014) C1orf163/RESA1 is a novel mitochondrial intermembrane space protein connected to respiratory chain assembly. *J Mol Biol* 426: 908-920
- Martinez Lyons A, Ardisson A, Reyes A, Robinson AJ, Moroni I, Ghezzi D, Fernandez-Vizarra E, Zeviani M (2016) COA7 (C1orf163/RESA1) mutations associated with mitochondrial leukoencephalopathy and cytochrome c oxidase deficiency. *J Med Genet* 53: 846-849
- Sullivan LM, Weinberg J, Keaney JF, Jr. (2016) Common Statistical Pitfalls in Basic Science Research. *J Am Heart Assoc* 5
- Tiranti V, Munaro M, Sandona D, Lamantea E, Rimoldi M, DiDonato S, Bisson R, Zeviani M (1995) Nuclear DNA origin of cytochrome c oxidase deficiency in Leigh's syndrome: genetic evidence based on patient's-derived rho degrees transformants. *Hum Mol Genet* 4: 2017-2023
- Topf U, Suppanz I, Samluk L, Wrobel L, Boser A, Sakowska P, Knapp B, Pietrzyk MK, Chacinska A, Warscheid B (2018) Quantitative proteomics identifies redox switches for global translation modulation by mitochondrially produced reactive oxygen species. *Nat Commun* 9: 324
- Wu WK, Volta V, Cho CH, Wu YC, Li HT, Yu L, Li ZJ, Sung JJ (2009) Repression of protein translation and mTOR signaling by proteasome inhibitor in colon cancer cells. *Biochem Biophys Res Commun* 386: 598-601

Referee #2 (Remarks for Author):

This manuscript characterizes COA7 (RESA1) as an IMS protein and a non-canonical substrate of MIA40. Some of the authors had previously reported a matrix localization of the protein but here its IMS residence and the involvement of MIA40 for its import are undoubtedly demonstrated. The protein is biomedically relevant because mutations in the human gene have been associated with mitochondrial leukoencephalopathy associated with mitochondrial respiratory chain complex IV deficiency. The authors identified the COA7 mutations as responsible for import failure, which leads to retention of the newly synthesized protein in the cytoplasm. However, the protein does not accumulate in this compartment because is actively degraded by the proteasome. The authors elegantly show that overexpression of mutant COA7 or inhibition of the proteasome with either MG132 or other clinically approved proteasome inhibitors restores COA7 levels in mitochondria and also its activity in complex IV assembly. Therefore, the authors suggest that proteasome inhibition may be a new venue to combat mitochondrial diseases associated with poor mitochondrial protein import or excessive degradation by the proteasome.

The manuscript is technically and conceptually sound, and appropriate for publication in EMBO Molecular Medicine.

I have only two requests to improve the manuscript:

1- The experiments presented do not completely exclude the possibility that mutant RESA1 is not degraded by mitochondrial proteases. The authors should silence some of these proteases (e.g. AAA proteases) and test whether the protein is still degraded with a similar efficiency.

A: In original manuscript we omitted the role of mitochondrial proteases in degradation of RESA1/COA7. In order to address this issue we performed silencing of YME1L in patient fibroblast. We observed a tendency towards increased levels of both mutated forms of RESA1/COA7 whenever YME1L was silenced (Fig EV3B). Silencing of YME1L was only partial so perhaps with more efficient approach we could observe more significant effects on RESA1/COA7 mutant levels. These data suggest that YME1L could be a protease, which degrades RESA1/COA7 in the IMS.

2- The authors suggest that clinically approved proteasome inhibitors, such as bortezomib, may be applied as therapeutic agents to combat at least a subset of mitochondrial disorders. The authors should discuss the potential side effects on mitochondria and other organelles.

A: We introduced into the discussion a comment on potential side effects of bortezomib treatment on mitochondria.

Minor points:

1- Page 10: "TIMM8A (CX9C) and COX19 (CX3C)" should be "TIMM8A (CX3C) and COX19 (CX9C)"

A: These mistakes were corrected in the revised manuscript.

2- I strongly suggest the authors to use COA7 to refer to the protein. The use of alternative names only serves to confuse the literature.

A: In the revised manuscript we refer to the protein as COA7.

Referee #3 (Remarks for Author):

This manuscript reports a careful analysis of the biogenesis of the mitochondrial respiratory chain assembly factor 1 (RESA1). RESA1 has been linked to mitochondrial leukoencephalopathy and complex IV deficiency. The authors show that RESA1, which contains 13 cysteine residues, is an unusual substrate of the mitochondrial intermembrane space assembly (MIA) system. In a remarkably complete characterization, they elucidated the molecular mechanisms of import of RESA1 into the mitochondrial intermembrane space, the interaction with Mia40 and disulfide bond formation. Importantly, the mitochondrial import of pathogenic mutant versions of RESA1 is slower than that of wild-type RESA1 and proteins accumulating in the cytosol are degraded by the proteasome. Using patient-derived fibroblasts, the authors discovered that inhibition of the proteasome rescued the localization of the mutant RESA1 to mitochondria and the activity of complex IV.

This paper by leading experts of the field is of technically very high quality and written very well. It provides exciting novel findings on the role of the proteasome in the pathogenesis of mitochondrial diseases and opens the way for new therapeutic approaches by using clinically approved proteasome inhibitors.

I have only a few minor comments on this exciting paper.

1. The authors provide a complete characterization of the biogenesis of wild-type and mutant RESA1 with important medical implications. It would be helpful for the general readership to present a cartoon of the import pathway of RESA1 and the role of the proteasome, e.g. in the last figure.

A: In the revised manuscript we included a schematic representation of the role of proteasome in mutant RESA1/COA7 degradation in Fig. 10.

2. Suggestions for corrections (indicated in CAPITAL letters):

- Page 3, middle: ... a precursor protein enters mitochondria via THE translocase of the outer membrane ...

- Page 8, line 3 from the bottom: These observations suggest that among the 13 cysteine residues, LIKELY 10 are involved in disulfide bonds, ...

- Page 17, middle: Treatment of patient fibroblasts with PROTEASOME inhibitors led to approximately ...

A: Page 3, middle – This sentence was omitted in the revised manuscript for sake of necessary manuscript shortening. We introduced other corrections in the revised manuscript.

2nd Editorial Decision

6 February 2019

Thank you for the submission of your revised manuscript to EMBO Molecular Medicine. We have now received the enclosed reports from the referees that were asked to re-assess it. As you will see the reviewers are now globally supportive and I am pleased to inform you that we will be able to accept your manuscript pending minor editorial amendments.

I look forward to reading a new revised version of your manuscript as soon as possible.

***** Reviewer's comments *****

Referee #1 (Remarks for Author):

The authors have addressed all comments in writing or experimentally. This is an exciting study that in my opinion is very well suited for EMBO Molecular Medicine.

Referee #2 (Remarks for Author):

The authors have now responded to all previous criticisms by this and other reviewers. The manuscript is now suitable for publication in EMBO Mol Med.

2nd Revision - authors' response

15 February 2019

Authors made the requested editorial changes.

Corresponding Author Name: Michal Wasilewski, Agnieszka Chacinska

Journal Submitted to: Embo Molecular Medicine

Manuscript Number: EMM-2018-09561